# Chronic microfiber exposure in adult Japanese medaka (*Oryzias latipes*)

**Lingling Hu**[1,2¤], **Melissa Chernick**[1], **Anna M. Lewis**[1], **P. Lee Ferguson**[1], **David E. Hinton**[1]*

1 Nicholas School of the Environment, Duke University, Durham, North Carolina, United States of America
2 State Key Laboratory of Estuarine and Coastal Research, East China Normal University, Shanghai, China

¤ Current address: College of Environment, Zhejiang University of Technology, Hangzhou, China
* dhinton@duke.edu

**Data Availability Statement:** Data are held in a public repository. The raw data set file is available from the Figshare database (doi: 10.6084/m9. figshare.10031471). All other data and figures are

## Abstract

Microplastic fibers (MFs) pollute aquatic habitats globally via sewage release, stormwater runoff, or atmospheric deposition. Of the synthetic MFs, polyester (PES) and polypropylene (PP) are the most common. Field studies show that fish ingest large quantities of MFs. However, few laboratory studies have addressed host responses, particularly at the organ and tissue levels. Adult Japanese medaka (*Oryzias latipes*), a laboratory model fish, were exposed to aqueous concentrations of PES or PP MFs (10,000 MFs/L) for 21 days. Medaka egested 1,367 ± 819 PES MFs (0.1 ± 0.04 mg) and 157 ± 105 PP MFs (1.4 ± 0.06 mg) per 24 hrs, with PP egestion increasing over time. Exposure did not result in changes in body condition, gonadosomatic- or hepatosomatic indices. PES exposure resulted in no reproductive changes, but females exposed to PP MFs produced more eggs over time. MF exposure did not affect embryonic mortality, development, or hatching. Scanning electron microscopy (SEM) of gills revealed denuding of epithelium on arches, fusion of primary lamellae, and increased mucus. Histologic sections revealed aneurysms in secondary lamellae, epithelial lifting, and swellings of inner opercular membrane that altered morphology of rostral most gill lamellae. SEM and histochemical analyses showed increased mucous cells and secretions on epithelium of foregut; however, overt abrasions with sloughing of cells were absent. For these reasons, increased focus at the tissue and cell levels proved necessary to appreciate toxicity associated with MFs.

## 1. Introduction

Microplastic pollution is a global environmental threat [1]. Microplastic fibers (to be referred to as microfibers; MFs) outnumber other types of microplastics, accounting for over 90% in some areas [2]. Worldwide, 9 million tons of fibers were produced in 2016, 60% of which were synthetics such as polyester, acrylic, polypropylene, and nylon [3]. The synthetic fibers used to make textiles (*e.g.*, clothing, upholstery, and rugs) shed MFs during washing and regular use; a single garment can shed over 1,900 MFs per wash [4]. MFs enter the aquatic environment via sewage release, stormwater runoff, or atmospheric deposition [3, 5–7] where they accumulate and impact biota [8]. Polyester (PES) and polypropylene (PP) are the most commonly used

in the manuscript and its Supporting Information file.

**Funding:** This work was supported by grants from China Scholarship Council ([2017]3109) to Lingling Hu. Sample prep and imaging for scanning electron microscopy was performed at the Duke University Shared Materials Instrumentation Facility (SMIF), a member of the North Carolina Research Triangle Nanotechnology Network (RTNN), which is supported by the National Science Foundation (Grant ECCS-1542015) as part of the National Nanotechnology Coordinated Infrastructure (NNCI).

**Competing interests:** NO authors have competing interests.

and most frequently observed synthetic MFs in the aquatic environment [3, 9]; hence, their selection for the present study.

Chemicals amended to textiles pose additional risks when released during laundering [10–12]. Studies of plastic leachates as well as effluents from textile industries have shown that dyes, surfactants, hydrocarbons, polymerizing monomers, and a variety of other compounds are released and negatively affect fish [13, 14]. For example, guppies (*Poecilia reticulata*) placed in textile dyeing effluent showed behavioral changes consistent with respiratory impairments including rapid opercular movements, gasping at the surface, and mucus thickening [15]. Gill histology revealed necrosis, hyperplasia, hypertrophy, lamellar fusion, increased mucus production, and sloughing of epithelium [15].

Field studies have reported MF ingestion in various species from zooplankton to mammals [16–19]. MFs have been detected in 60% of macroinvertebrates, 49% of shorebirds [16] and in a variety of fishes [20–22]. For example, Halstead et al. [20] studied fish from an urbanized estuary in the northern arm of Sydney Harbor, Australia and found PES MFs made up the majority (83%) of microplastic contents in gut lumens.

Despite MFs making up the highest percentage of plastics in specimens collected from the field, there are few laboratory studies describing effects, particularly in fish. Grigorakis et al. [23] found that MFs (50–500 μm long) amended to goldfish (*Carassius auratus*) diet did not remain in gut any longer than other dietary components. Goldfish fed food containing ethylene vinyl acetate (EVA) MFs (0.7–5.0 mm long) for 6 weeks exhibited damage to the buccal cavity including abrasions to epithelium as well as damage to gill filaments and folds of the gut [24].

Gill and gut are sensitive targets for pollutants due to their large surface area and intimate interface with the external environment [25, 26]. Gill alterations can impact vital physiological processes, including: ionic balance, acid-base equilibrium, gaseous exchange, excretion of nitrogenous wastes, and osmoregulation [27]. Although it is a primary barrier to the external environment, less is known about effects on fish intestinal mucosa [26]. And while MFs are often reported in these sites upon necropsy, there are rare descriptions of tissue alterations.

Japanese medaka (*Oryzias latipes*) are a well-established aquarium model fish [28] that are small in size, agastric, have daily oviposition, are easily cultured, have characteristic developmental stages [29–32], and a well-defined anatomy [33, 34]. With these characteristics in mind, we sought to determine chronic effects of MFs in this model using exposures with controlled number, type, and characteristics of MFs that increased precision in determination of host responses.

## 2. Materials and methods

### 2.1 Experimental animals

Our colony of orange-red (OR) medaka is maintained at Duke University under protocols approved by the Duke University Institutional Animal Care and Use Committee (IACUC). Adult, brood stock medaka were maintained at 24°C with a pH of 7.4 in an AHAB recirculating system (Pentair Aquatic Eco-Systems, Apopka, FL) and a 14:10 light:dark cycle. Otohime β1 commercial dry diet (200–360 μm, Pentair Aquatic Eco-Systems) was fed to fish three times per day, and *Artemia* nauplii (90% Great Lakes Strain, Pentair Aquatic Eco-Systems) were fed along with dry diet during the morning and afternoon feedings.

### 2.2 Microfibers

Commercially dyed green polyester thread (PES, 10–20 μm diameter) and transparent polypropylene fibers (PP, 50–60 μm diameter) were purchased from a supermarket (Shanghai

Qinhe, China). Polymers were verified using a micro-Fourier Transformed Infrared spectroscopy microscope (LUMOS µ-FT-IR, Bruker, Beijing, China) in attenuated total reflectance (ATR) mode (Fig 1A$_1$–1B$_1$) [9]. Next, strands were cut crosswise with clean micro-scissors

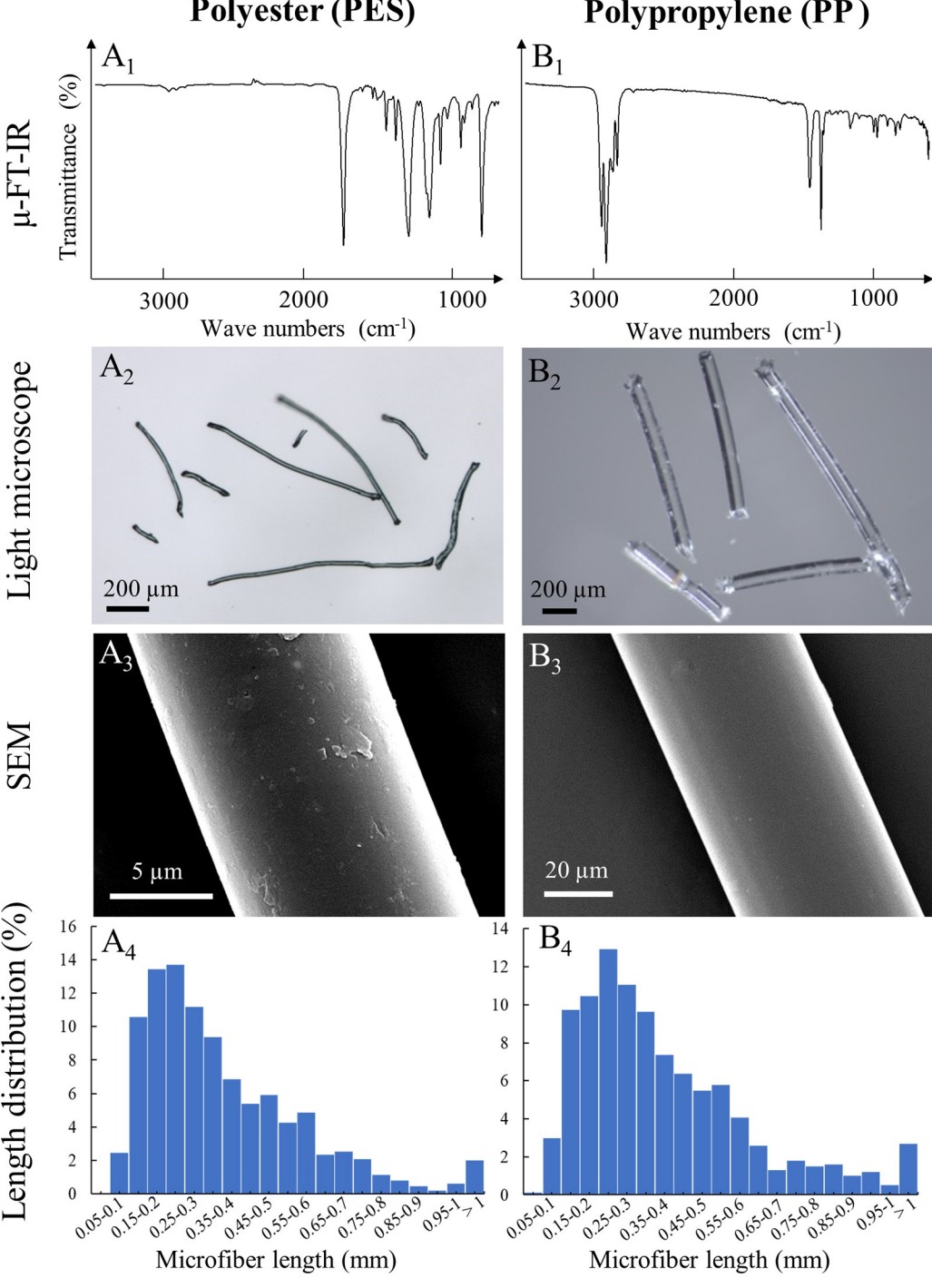

**Fig 1. Characteristics of PES and PP microfibers.** Column A pertains to green PES MFs while column B is transparent PP MFs. µ-FT-IR characterized and confirmed polymers in each MF type (A$_1$ and B$_1$). Brightfield images are shown in A$_2$ and B$_2$. Surface structure imaged with SEM is in A$_3$ and B$_3$. Size distributions show PES fibers averaged 350 µm in length (A$_4$) and PP averaged 380 µm long (B$_4$).

(Ted Pella, Redding, CA) into MFs and stored in a clean glass bottle until use. Surface features were imaged by scanning electron microscopy (FEI XL30 SEM-FEG, Thermo Fisher Scientific, Waltham, MA) (Fig 1A₃–1B₃).

A standard curve was produced in order to determine number by mass (mg dry weight (dw)) for each MF type (S1B and S1D Fig, S1 Table). In this way MFs could be weighed and then added directly to tanks during water changes to yield selected concentrations, providing a practical method for MF addition while avoiding loss of MFs that would occur during transfer in a liquid medium. To make the regressions, five gradient masses of each MF type were weighed (mg dw) and then soaked in 10 mL 70% ethanol (EtOH) to disperse (S1A and S1C Fig). Each MF suspension was mixed using a glass Pasteur pipette and 1 mL was filtered through a polycarbonate membrane filter (Millipore TMTP04700, 47 mm diameter and 5 μm pore size) under vacuum and imaged using a Nikon SMZ 1500 stereomicroscope with a Nikon DXM1200 camera and Nikon NIS-Elements 3.10 software (Nikon Instruments Inc., Melville, NY). MFs were counted in three non-overlapping images, and the total number of MFs by weight (mg dw) was calculated. The filtering, imaging, and counting procedures were repeated in triplicate. A linear regression was used to establish mass vs. number of MFs (S1B and S1D Fig).

MF size distribution was determined by measuring the length of approximately 1,000 individual fibers using 50 randomly selected images and ImageJ 1.48 software [35]. 98.0% of PES MFs were < 1000 μm in length and 78.8% were < 500 μm, with an average length of 350 μm (Fig 1A₄). 97.3% of PP MFs were < 1000 μm in length and 76.0% were < 500 μm (Fig 1B₄), with an average length of 380 μm.

## 2.3 Preliminary study

A preliminary study was conducted to determine 1) whether aqueous exposures to MFs would result in uptake and 2) how and in what quantities MFs should be used. Adult, 8-month old medaka were randomly selected from our colony. Eight breeding pairs (1 male, 1 female) were placed in 3 L tanks containing 2 L of batch water (0.1% w/v artificial salt (Instant Ocean, Blacksburg, VA) in MilliQ water (Millipore Sigma, Burlington, MA)) that had been mixed and oxygenated with an air stone for at least 12 h prior to use. Tanks were maintained in a dedicated room at 24˚C and under a 14:10 light:dark cycle. Fish were acclimated to these conditions for three days. Then, air stones (Saim's Store, Amazon.com; 14.5×25 mm) were added to each tank and fish further acclimated for three days. In addition to oxygenation, air stones kept MFs mixed and suspended upon addition to tanks. Fish were fed two times per day with 1% body weight Otohime β1 and an equal amount of *Artemia* nauplii culture. Along with a control, the following concentrations of MFs were tested: 1,000 fibers/L PP, 1,000 fibers/L PES, and 10,000 fibers/L PES. 10,000 fibers/L was chosen as an upper limit based on levels detected in Arctic sea ice (12,000 ± 14,000 particles/L) [36], a laboratory study with zebrafish (*Danio rerio*) [37], and projected increases in the environment [38]. Each treatment had 2 replicate tanks (2 breeding pairs/treatment), with an exposure time of 21 days.

During feeding times, air stones were temporarily inactivated to ensure that dry food remained on the water surface to allow fish to feed in their accustomed manner. Any MFs stuck to tank walls during the static period were resuspended with a pipette, then food was introduced. Fish were allowed to feed for 5 mins before aeration was resumed. We did not observe preferential binding of MFs to food. Fish were routinely observed daily during feeding- and non-feeding times for alterations in normal behavior (*e.g.*, increased opercular movements, erratic swimming, piping, cowering) that might indicate stress. Eggs were removed from tanks daily by siphoning bottoms of tanks, cleaned and assessed as described below

(section 2.5). No changes in fecundity were found among different groups. Every 2–3 days, tanks were siphoned to remove feces and 25% (500 mL) water was removed and replaced with clean batch water. Then, new dry MFs were added by mass to replace those removed using the generated standard curve (S1 Fig). On days 6, 13, and 20, a complete (100%) water change was done, and tanks and air stones were thoroughly cleaned.

Fecal material was collected 24 hrs after a complete water change using a 7.5 mL transfer pipette (VWR) to minimize removal of MF-containing tank water. We did not observe preferential binding of suspended MFs to feces when observed using a stereomicroscope (Nikon SMZ1500). Feces were placed into pre-weighed 1.7 mL Eppendorf tubes (1 tube/tank), centrifuged for 5 min at 5,000 rcf, and supernatant removed. Feces were then digested using hydrogen peroxide ($H_2O_2$; 30%, v/v, JT Baker, Avantor, Allentown, PA) at 65ºC for 6 h and then filtered (Millipore TMTP04700) under vacuum, digitally imaged, and counted as described above (section 2.2). The lower concentrations of MFs (1,000/L) had an average of 22.5 PP and 20.5 PES per fish per day, while fish exposed to 10,000 fibers/L had an average of 1002.9 items per fish per day. Accordingly, the higher concentration was selected for use in the definitive study.

After 24 hrs and after 21 d, a single male from each treatment group was euthanized by immersion in an ice water bath (i.e., rapid cooling) until vital signs (e.g., opercular movement, righting equilibrium, fin and muscle movement, and heartbeat) had ceased [39, 40]. Then gill, gut and liver were removed. Excised organs were individually digested using $H_2O_2$ (30% v/v) at 65ºC for 24 hrs. Resultant digestates were filtered (Millipore TMTP04700) under vacuum and then examined under a stereomicroscope. PP and PES MFs were restricted to gut digestates.

## 2.4 Experimental design

Thirty-three breeding pairs, randomly selected from our colony, were moved to the dedicated room (section 2.3) and first evaluated for reproductive status and fecundity by observations over 7 consecutive days; resultant embryos were counted and assessed for normal development and viability. Twenty-seven pairs with the highest and most consistent productivity (e.g., same number of eggs each day) were randomly assigned to treatment groups (control, PES, or PP), with 9 replicate pairs per group. There were no significant differences in egg production and fertilization rate for each group before exposure (Fig 2). Fish were placed in tanks with air stones, acclimated, and fed as described above (section 2.3), followed by the addition of 10,000 fibers/L to each tank (S1 Fig, S1 Table). Exposure duration was 21 days, during which feeding, water changes, and embryo collection followed methods used in the preliminary study. Based on results of the preliminary study, an additional 1,000 MFs per fish per day were added during water replacements to account for MFs bound and removed in fecal material. All individuals were weighed (mg wet weight (ww)) before exposure and once weekly during the experiment. Tank water samples were taken immediately before complete water changes, filtered (0.2μm) to remove MFs, and stored at -80ºC for future chemical analyses to determine presence and concentrations of dyes and other additives.

## 2.5 Embryo and fecal collection and analyses

At 7, 14, and 21 days, feces were collected by siphoning bottoms of each tank, imaged under a stereomicroscope and then transferred to pre-weighed 1.7 mL Eppendorf tubes (1 tube/tank). Samples were processed with $H_2O_2$ and counted as described above (section 2.3), enabling calculation of MF number and length.

Eggs were collected within 24 h after complete water changes. Before and after feedings, and every 2–3 hrs, deposited eggs were collected with a 7.5 mL transfer pipette. Then, clutches

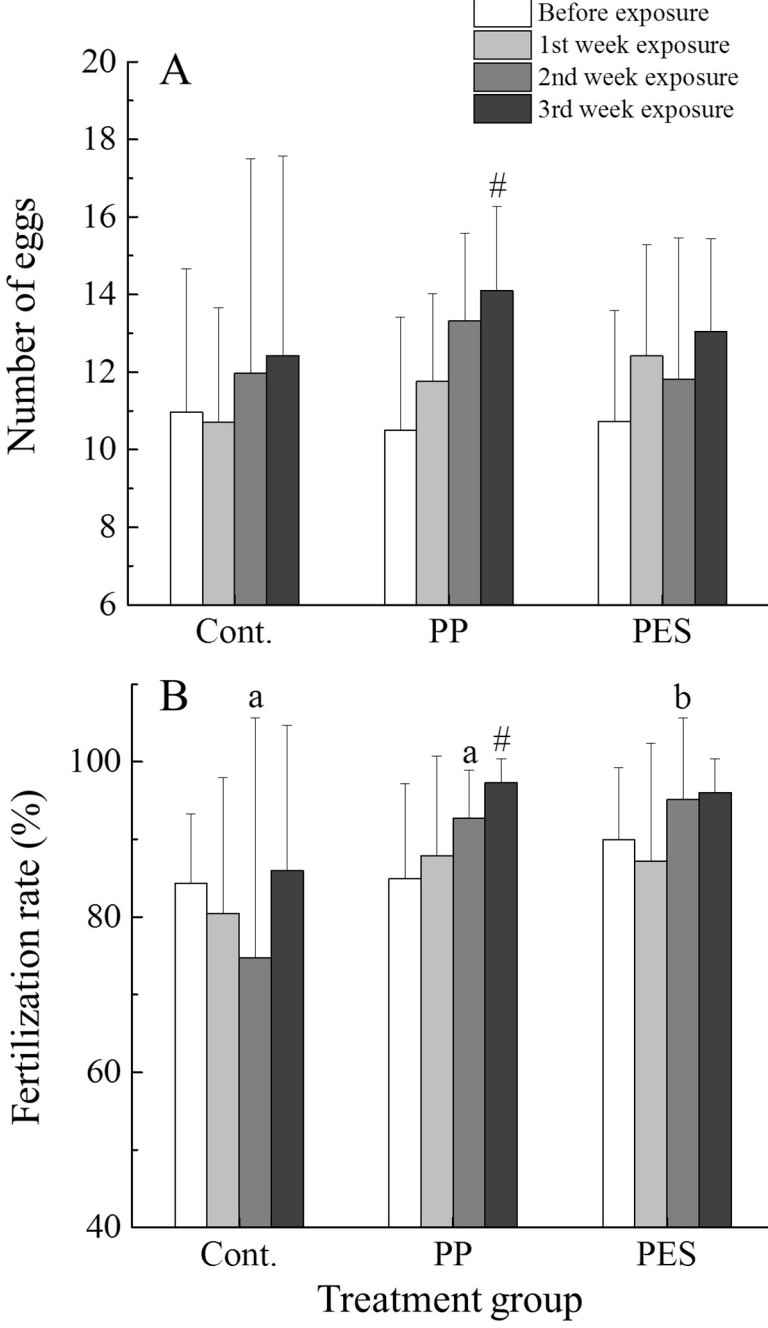

**Fig 2.** Egg production (A) and fertilization rate (B) for control (Cont.), polypropylene (PP), and polyester (PES) MFs groups before and throughout the course of the exposure period. Bars represent means ±SD (n = 9 pairs). Different letters indicate significant differences in fertilization rate (%) comparing time points between different treatment groups, p < 0.05. Pound symbols (#) indicate significant differences (p < 0.05) between time points within a treatment group.

were rolled on moistened paper towels to separate and clean individual eggs before they were transferred to labeled Petri dishes (VWR, Corning) containing batch water [31]. Next, counts were made of fertilized embryos, unfertilized eggs, and non-viable embryos. The latter two were then discarded. Embryos were maintained on an orbital shaker at 60 rpm (Thermo Fisher

Scientific) in an incubator at 26˚C with a 14:10 light:dark cycle. Embryo mortality, hatching, and development were observed daily [31, 32]. At 14 d post fertilization (dpf), each larva was anesthetized in 150 mg/L tricaine methanesulfonate (MS-222; Sigma-Aldrich, St. Louis, MO) and total body length was measured.

## 2.6 Fish sample preparation

After 21 d of exposure, all fish were euthanized via rapid cooling, imaged (Panasonic, HC-X920, Newark, NJ), and weighed (mg ww). Standard length (mm) and girth at pectoral girdle (mm) were measured for each fish using ImageJ. Three breeding pairs (n = 6) from each treatment group were allocated for histology and a ventral midline incision was made from anus to near the pectoral girdle. Then, 10% neutral buffered formalin (10% NBF; VWR) was flushed through the incision using a disposable transfer pipette with extended fine tip (VWR). The pipette was also inserted into the buccal cavity to gently perfuse fixative into buccal cavity, pharynx, branchial cavity and esophagus. This flushing facilitated fixation of deeper tissues. These specimens were placed in 50-mL conical tubes (1 pair/tube) filled with 10% NBF and fixed at room temperature overnight. Then these tubes were moved to 4ºC for storage until time of processing for histology (section 2.7).

Fish of the remaining 6 breeding pairs (n = 12) were dissected, and gill, gut, gonads, and liver excised and weighed (mg ww). For 3 of these pairs (n = 6), gills and gut were fixed for SEM (section 2.8). For the remaining 3 pairs (n = 6), all excised organs were frozen and stored at -80ºC for future chemical analyses of additives.

To evaluate condition of individual fish, the following indices were calculated: coefficient of condition ($K$), gonadosomatic index (GSI), and hepatosomatic index (HSI) [41, 42] using the following formulae:

$$K = \frac{100,000 \ W_{body}}{L^3} \tag{1}$$

$$GSI = \frac{W_{gonad}}{W_{body}} \times 100 \tag{2}$$

$$HSI = \frac{W_{liver}}{W_{body}} \times 100 \tag{3}$$

*where*: $W_{body}$ = body weight (g ww); $L$ = standard length (mm); $W_{gonad}$ = gonad (ovaries or testes) weight (g ww); and $W_{liver}$ = liver weight (g ww).

## 2.7 Light microscopic analysis

Fixed specimens were processed, sectioned and stained at the Histology Laboratory, Department of Population, Health and Pathobiology, North Carolina State University College of Veterinary Medicine, Raleigh, N.C. First, fixed specimens were decalcified in 10% formic acid for 48 h and washed with water. In an automated tissue processor (Thermo Shandon Path Centre, Grand Island, NY), a graded series of EtOH solutions was used for dehydration and then Clear-Rite 3 (Richard Allen Scientific, Kalamazoo, MI) used for clearing. Specimens were then embedded in paraffin and oriented in left lateral recumbency, with one breeding pair in each block. 5 μm thick, step sections were cut with a Leica 2135 rotary microtome (Leica Biosystems Inc., Buffalo Grove, IL) and mounted on glass histological slides. Such orientation and embedment assured an average of 8 sections per pair of fish, yielding repeated views of each major organ in parasagittal planes. Hematoxylin and eosin (H&E) stained sections were used

for general tissue survey. Alcian Blue and Periodic Acid Schiff (AB-PAS) stained one slide per pair for mucus. All slides were examined and imaged with a compound light microscope (Nikon E600, Nikon Instruments, Inc., Melville, NY). The Penn State Zebrafish Bio-Atlas [43] was consulted and used for comparison.

Because fewer than expected MFs were observed during dissections or in histologic sections of branchial chamber and gill structures, we conducted an additional exposure to determine passage of MFs through branchial chambers. One new pair per treatment was placed in tanks and acclimated as described above (section 2.3). Fish were exposed to 10,000 fibers/L for 48 hrs, euthanized via rapid cooling, and then the operculum was carefully removed to image underlying gill filaments with a stereomicroscope. Next, approximately 2 mL fluid containing 10,000 fibers/L was gently flushed into the buccal cavity near the oral flaps and observations made of their passage through or around gills. Finally, gills were removed and imaged under a stereomicroscope (Fig 5A and 5B).

## 2.8 Scanning electron microscopic analysis

All SEM fixation and processing were adapted from published procedures [44]. Briefly, gills (arches with rakers and filaments attached) and gut were fixed overnight at 4ºC in 2.5% glutaraldehyde (Electron Microscopy Sciences, Hatfield, PA) buffered with a cacodylate-sucrose solution (0.1 mol L$^{-1}$ sodium cacodylate and 0.1 mol L$^{-1}$ sucrose, pH 7.6). Using a sterile, single-edged razor blade, transverse sections (2–4 mm) were cut from the fore-, mid-, and hindgut of each fish in order to visualize mucosal surfaces of folds. Just prior to preparation for SEM, samples were washed in 0.1 M phosphate buffered-sucrose solution for 20 min and dehydrated using an EtOH series (30%, 50%, 70%, 90%, 100%, 100%; 15 min each). Organs were then transferred through an amyl acetate (Electron Microscopy Sciences) series (amyl acetate: EtOH::1:3, amyl acetate:EtOH:: 3:1 and then two changes of l00% amyl acetate; 15 min each). Gills were critical point dried (LADD, Williston, VT) and gut samples were dried with hexamethyldisilazane (HMDS, Electron Microscopy Sciences; three changes of l00% HMDS, 10 min each).

Processed samples were placed on carbon tape (Electron Microscopy Sciences) affixed to a pin stub (12.7 × 8 mm, Ted Pella Redding, CA) and sputter-coated with gold using a Denton Desk IV (Denton Vacuum, Moorestown, NJ). To reduce charging from settling of gold, gut samples were sputter coated immediately prior to imaging. All samples were observed using a SEM with a spot size of 3 and an accelerating voltage of 15–20 kV and imaged with Scandium software (ResAlta, Golden, CO).

## 2.9 Statistical analyses

Statistical analyses were performed using SPSS 22.0 (IBM Armonk, NY) software and Origin 9.0 (OriginLab Corporation, Northampton, MA) software. Kolmogorov-Smirnov and Shapiro-Wilk tests were performed to test for normality, and a Levene test was used for homogeneity of variance. Data were not normally distributed and had unequal variance; therefore, non-parametric tests were used. Data for quantities of MFs, number of embryos, adult body weight, and larval body length had factors for time as well as treatment. Therefore, a Mann-Whitney $U$-test was used to determine differences between time points within a treatment group, and a Wilcoxon test was used to test differences between treatment groups within a time point. The Wilcoxon test was also used to determine differences in $K$ (n = 18), GSI (n = 6 females, 6 males), HSI (n = 6 females, 6 males), adult body weight before and after exposure, embryo mortality, hatching, and developmental endpoints. A $p < 0.05$ was considered statistically significant.

## 3. Results

### 3.1 Medaka condition

All fish survived the exposure period and no change in body weight of female fish occurred. Body weight of males in all groups was significantly increased (S2B Fig, p = 0.011 for control and PES-exposed males, p = 0.008 for PP-exposed males). There were no significant differences in fish condition assessment indices of either sex including *K*, HSI, or GSI among treatment groups (S2 Table).

### 3.2 Fecundity and embryo development

Egg production and fertilization success following the first week of exposure did not differ from results prior to exposure (Fig 2). During the second week of exposure, fertilization rate in PES-exposed pairs was greater than other groups (Fig 2B). Females exposed to PP MFs produced more eggs over the course of the experiment, becoming significantly higher than before exposure values by the last week (Fig 2A, p = 0.013). Their mates were able to successfully fertilize this greater number of eggs (Fig 2B, p = 0.017). There were no statistical differences in mortality, development, or hatching success for embryos collected at days 7, 14 or 21 compared to controls (S3 and S4 Figs). Additionally, body lengths of larvae after hatch were the same between control and treatment groups (S5 Fig). These results were consistent with- or better than those observed in routine repeated assessments of our breeding colony.

### 3.3 Fecal MFs abundance

No MFs were found in feces of control fish. MF-laden feces in exposed fish provided quantitative evidence of ingestion and egestion (Fig 3). The abundance of PP MFs ranged from 23 to 447 items per fish per day (average: 157 ± 105 items/fish/day). Interestingly, PP MF numerical density was significantly higher at day 21 (p = 0.031 for day 7 vs. day 21, p = 0.042 for day 14 vs. day 21; Fig 3D). PES MF abundance ranged from 340 to 3097 items/fish/day (average: 1367 ± 819 items/fish/day, Fig 3D). Excretion of PES MFs was significantly greater than that of all PP MFs (p<0.001, Fig 3D) but did not change over time.

### 3.4 Histological changes

Light micrographs of gills from individuals exposed to PES MFs for 48 hrs showed aneurysms along lamellae (Fig 4B). We also observed that MFs were able to pass through branchial chamber but did not become entangled in gill filaments (Fig 4C and S6G Fig). This finding was in line with that observed in histological sections (Fig 4E) after 21 d of exposure. PES MFs were present in buccal cavity, on pharyngeal mucosa near teeth, in branchial cavity, and on gut folds (S6D–S6H Fig). Green dye facilitated recognition of PES MFs in sections, while PP MFs were only identified in AB-PAS stained sections as negatively stained, clear spaces (S6I Fig) identical in diameter to PP MFs observed in initial MF characterizations.

Medaka branchial cavity and gills showed alterations upon exposure to PES and PP MFs. The wall of branchial cavity covering medial aspect of operculae presented as a rounded balloon-shaped structure under low magnification (Fig 4F). This altered inner opercular membrane appeared to push against gill filaments resulting in deformation of the most rostral primary and secondary lamellae (Fig 4F). This swelling occurred in half of the fish in each MF treatment. An additional site of swelling was beneath the epithelium of the caudal wall of the branchial chamber (Fig 4F). While not as large, it also made contact with primary lamellae. Other alterations were gill specific including aneurysms, epithelial lifting with separation from underlying structures in inter-secondary lamellar spaces, partial and complete lamellar fusion,

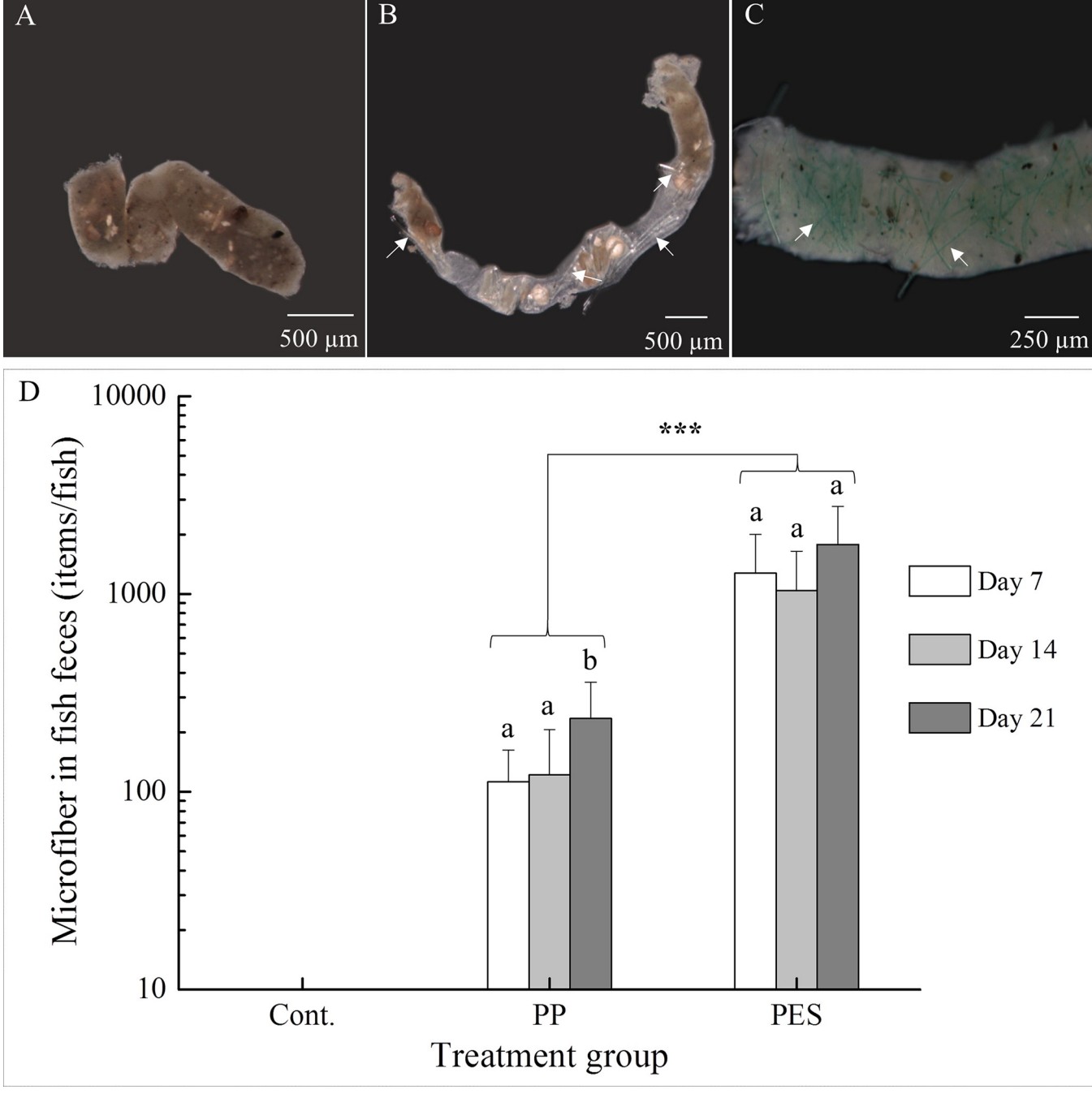

**Fig 3. Egestion of MFs.** Images of feces from medaka exposed to control (A), transparent PP MFs (B), or green PES MFs (C). White arrows point to fibers. MF abundances on day 7, 14, and 21 are represented in the histogram (D). MF abundances are expressed as mean ±SD (n = 9 pairs). Different letters indicate significant differences in MF abundances between time points within a treatment group (Wilcoxon test, p < 0.05). Asterisks (***) indicate significant difference between treatment groups (Mann-Whitney *U*-test, p < 0.001).

and erosion of epithelium from secondary lamellae (Fig 4E–4I). Petechiae (*i.e.*, small spots of hemorrhage) and epithelial lifting were found in gills of 50% of control fish, but were minor in size and extent, with rare petechiae in different positions along the gill filament. Conversely, aneurysms and epithelial lifting occurred in gills of 67% of PES-treated and 83% of PP-treated

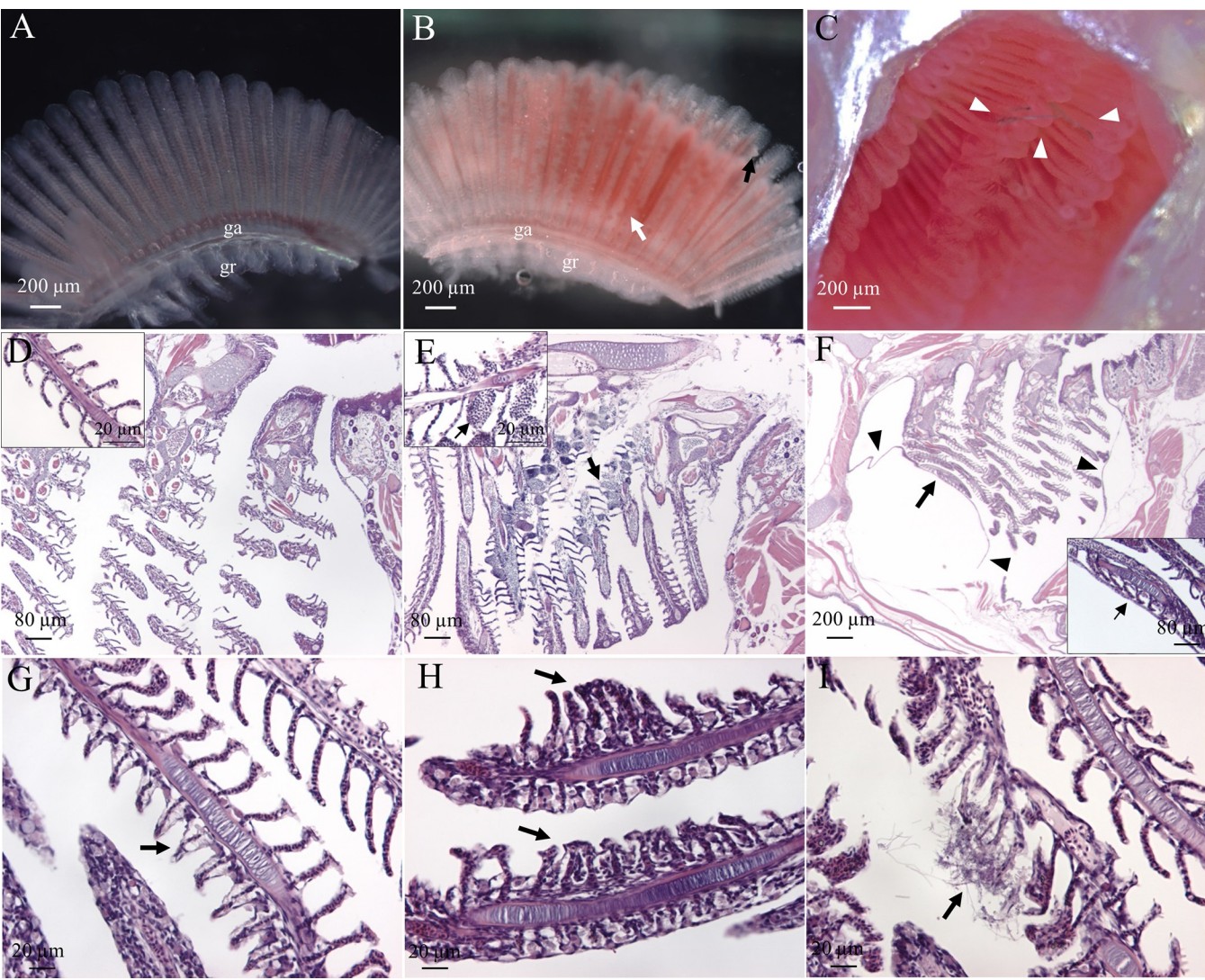

**Fig 4. Gill alterations following MF exposure.** Light micrographs of gills from control (A) and PES-exposed (B) medaka after 2 days of exposure; white arrow indicates aneurysms, black arrow indicates normal lamellar outgrowths, arrowheads indicate PES MFs in the branchial cavity (C). H&E stained histological sections of gills from the adult medaka exposed to 0 (D), PES (E-G, I), or PP (H) MFs for 21 days. (E) Black arrows under low and high magnifications of the filaments indicate aneurysms. (F) Arrowheads indicate swelling between deep layers of the operculum associated with the wall of the branchial chamber and arrows show pushing of inner opercular epithelium against gill primary lamellae, visible in more detail in high magnification inset. (G) Arrow indicates epithelial lifting in the secondary lamellae. (H) Arrows indicate fusion of secondary lamellae. (I) Arrow indicates epithelial alterations of the secondary lamellae. ga, gill arch; gr, gill raker.

fish and were numerous and mainly concentrated along water outflow tracts (*i.e.*, passages between adjacent gill arches and their associated primary lamellae) (Fig 4E). Fusion of secondary lamellae did not occur in controls, but was observed in MF-treated fish, most frequently (67%) after PP exposure and less so after PES exposure (33%) (Figs 4H and 5H).

H&E staining showed no alterations in internal organs (liver, kidney, thyroid, heart, spleen, pancreas, and gonads) of exposed individuals. AB-PAS stained sections of control revealed mucus in gut lumen and in goblet cells (S7A and S7B Fig). Both PES and PP groups revealed large amounts of mucus in foregut lumen and numerous, enlarged goblet cells (S7C–S7F Fig). Such alterations were absent in mid- and hindgut. No evidence was seen for abrasions, erosion, or other alterations in any segment of gut.

## 3.5 Surficial observations

SEM of control gills showed intact filaments with uniform inter-lamellar spaces (Fig 5A), smooth and intact surfaces of gill arches and rakers (Fig 5B), and mucous cells with minimal mucus production (Fig 5C). PES-exposed fish exhibited surface erosion of gill filaments and arches (Fig 5D and 5E). Primary lamellar tips were fused and enhanced terminal outgrowths of secondary lamellae were seen in one of three PP-exposed fish (Fig 5H). In both treatment groups, increased mucous production was observed as strands and sheets over filaments (Fig 5D and 5G) and rakers (Fig 5E). Increased output from individual mucous cells (Fig 5F, 5I) was also observed in both treatment groups.

SEM of control gut revealed regular, elongated enterocytes and pores for mucus secretion in and on folds of fore-, mid-, and hindgut (Fig 6A and 6B). Increased mucus was observed in foregut of PES exposed fish, but no other changes were seen (Fig 6C). Rarely, PES MFs were found trapped in the folds (Fig 6D), but most MFs were oriented longitudinally and were encased in food, mucus, and waste materials within the lumen (Fig 6E). Interestingly, grooves,

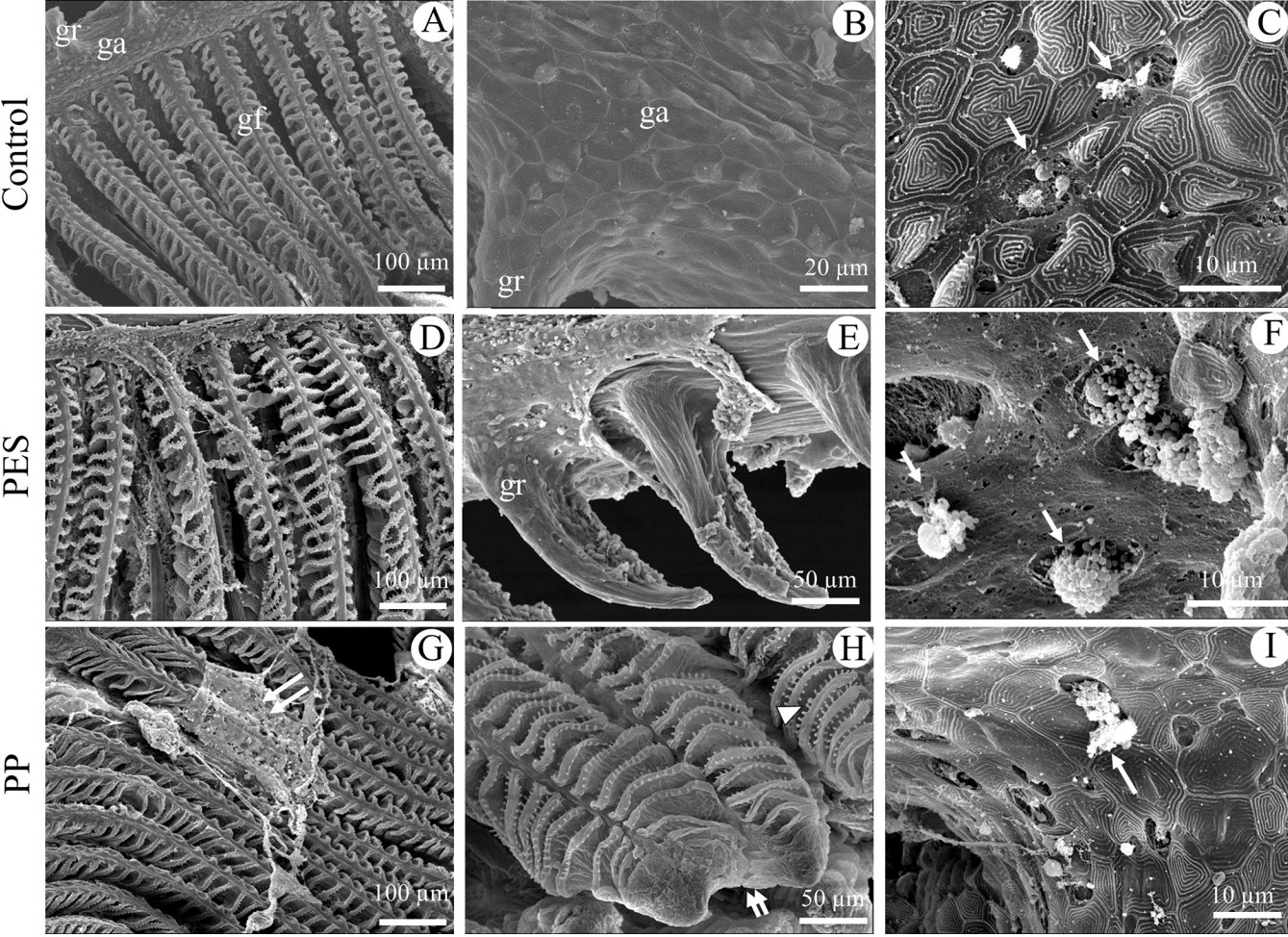

**Fig 5. SEM of gills after 21 days of exposure.** 0 (Control; A-C), PES (D-F), or PP (G-I) MFs. (A, D, G) gill filaments, only a portion of one gill raker may be observed in control figure (A). (B) Gill arch. (E) Gill raker. (C, F, I) Magnification of gill arch showing mucous cells indicated by white arrows. (G) Double white arrow indicates mucous secretion as a sheet; (H) Magnification of the filament tip with arrowhead to outgrowth and showing fusion of distal tips of adjacent primary lamellae (double white arrow). ga, gill arch; gf, gill filament; gr, gill raker.

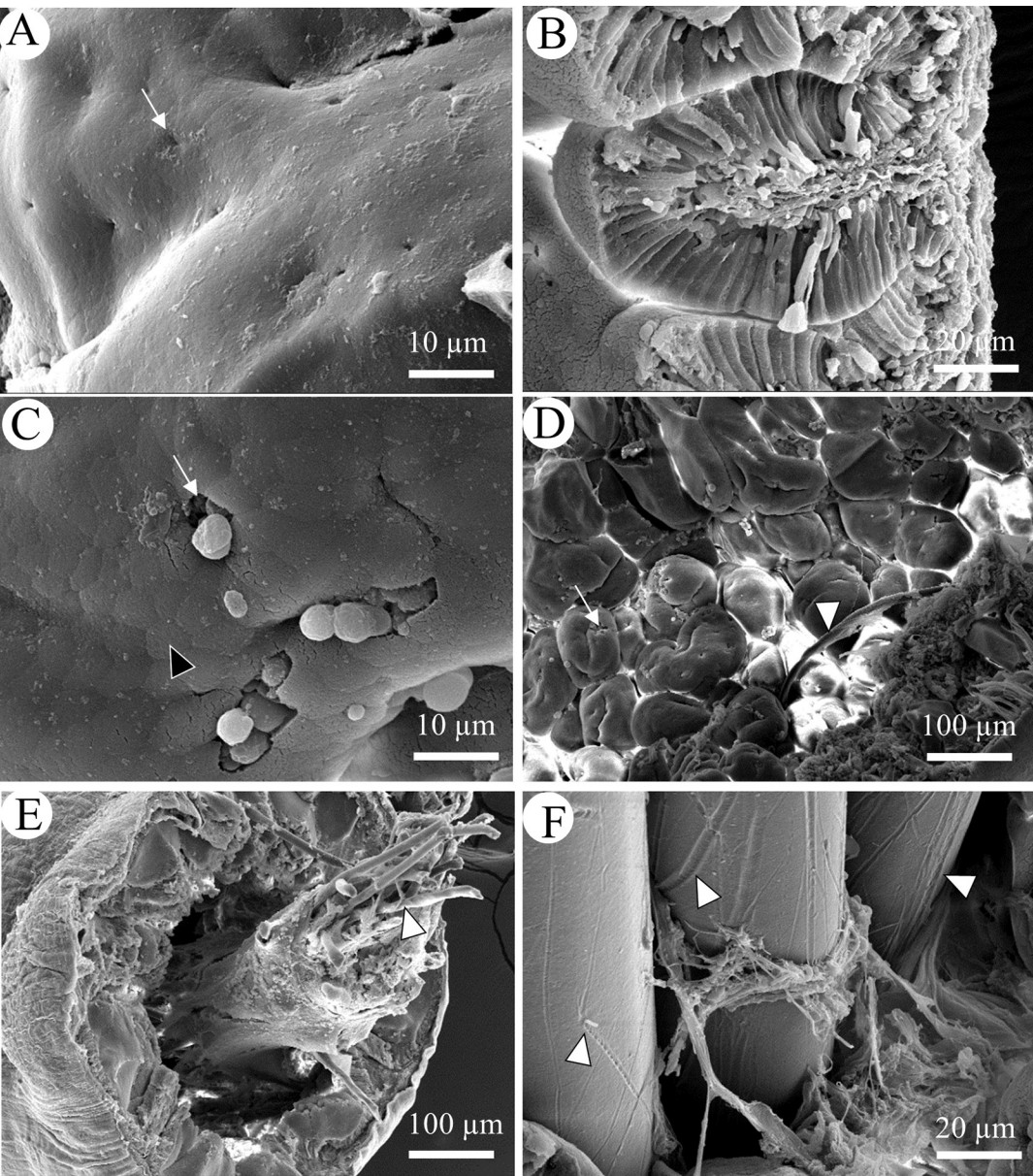

**Fig 6. SEM of cross sections of gut.** (A) Surface epithelium of foregut from a control fish with white arrows marking pores for mucous secretion; (B) Transverse section of hindgut from control fish; (C) Surface epithelium of foregut from PES-exposed fish, black arrowhead indicates apical tips of enterocyte, white arrow indicates mucus secretion; (D) Low magnification of foregut from PES-exposed fish with fiber entangled in folds (white arrowhead); (E) Low magnification of hindgut from PES-exposed fish with fibers (white arrowhead) encased in digesta; (F) High magnification of PP in hindgut showing elongated grooves on their surfaces.

that were not observed in pristine fibers (Fig 1B3), were found on surfaces of PP MFs in the hindgut (Fig 6F).

## 4. Discussion

This study addressed chronic effects of two types of MFs on adult medaka under controlled laboratory conditions. A thorough assessment was made of MF entry, egress, and interaction

with tissues as they passed through head gut, branchial chamber, and digestive system. While there are reports for several types of plastics associated with the above sites, there are little to no detailed assessments with respect to MFs.

## 4.1 Body condition

MFs exposure did not affect medaka body condition or indices over the 21 d suggesting no decreases in food intake or nutrition. Body weight of males in all groups, including control, increased without corresponding increases in *K*, suggesting males grew larger overall. Because breeding pairs were housed in relatively large tanks with ample diet, it is possible that males were less active in that they did not have to compete for females. Growth and weight as end-points of microplastics exposure vary, with some fish showing reductions [e.g., 24, 45, 46] and others no changes [e.g., 37, 47]. As might be expected, this variation seems to be the result of several factors including species, life stage, exposure duration, microplastic size and polymer.

## 4.2 Fecundity

The effects of microplastics on reproduction have been investigated in various invertebrate species such as oysters, water fleas, and cnidarians [48–50]. Such studies typically report decreased reproductive output (*e.g.*, oocyte number, fertilization rate) [49, 50]. However, little data exists on reproductive effects of microplastics in fish. In our study, medaka exposed to PP MFs had a significant increase in egg production and associated fertilization rate over time. Changes in egg number are a common biomarker of endocrine disruption in fish [51, 52]. Studies of single plasticizers have reported biological effects at ng/L or μg/L concentrations [53], and even low doses can disrupt endocrine systems [54]. No increases were observed in control fish and the MFs did not leave the digestive tract. Therefore, it is plausible that additives leached from MFs in the digestive tract and/or while in the water column. Turbulence such as that created by the air stones for MF mixing in the present study may have increased this additive leaching in water [55].

A hazardous substance that remains within plastic has a lower risk; it needs to be leached/released/desorbed for toxicity to occur [56]. This can occur in all phases of a plastic's life cycle, in a variety of media, and can depend on the composition of non-polymeric substances [56]. However, determination of type and magnitude of leaching is complex as it depends on a multitude of factors [56]. There is also a lack of data about the actual content of additives in textiles in the common market, primarily due to difficulties in obtaining information from producers on substances used during manufacturing [11].

Rochman et al. [57] exposed adult medaka to polyethylene (PE) microplastics and found changes in estrogen receptor mediated gene expression and altered testicular histopathology, suggesting endocrine system function was affected. In marine medaka (*Oryzias melastigma*), the additive di-(2-ethylhexyl)-phthalate (DEHP) and its active metabolite mono(2-ethylhexyl)-phthalate (MEHP) disrupted endocrine function and accelerated spawning start time and decreased fecundity in a sex-specific manner [58]. In contrast, we observed an increase in female fecundity upon exposure to PP MFs. Various additives (e.g., bisphenol A (BPA)) have been shown to produce estrogenic effects, including the induction of vitellogenin and may be an androgen receptor agonist [see review in 59]. Benzotriazoles (BTris), abundant in clothing textiles, are persistent in the environment and are known to have bioaccumulative properties [11]. Following aqueous exposure to BTris (0.01–1 mg/L) for 4 or 35 days, adult marine medaka had induced vitellogenin (VTG) gene expression in liver, gills, and gut of both sexes, down-regulated CYP1A1 gene expression levels in liver and gut, and induced CYP19a expression in ovaries [60]. Those results indicate BTris is an endocrine disruptor in that VTG

production is estrogen dependent, many estrogenic chemicals have been reported to inhibit CYP1A1, and CYP19a is involved the control of various physiological functions of estrogens [60].The pristine MFs, stored tissues and tank water from our study are currently undergoing chemical analysis to assess the extent to which leaching may have occurred. Only after this analysis will we be able to directly link effects to specific chemicals.

It should be noted that most exposure studies have used pristine microspheres or fragments. There are knowledge gaps as to how MFs behave in the environment [61]. Several dyes and chemicals used in the manufacture of textiles have been shown to be acutely toxic [13, 15] or carcinogenic [62]. The ability of plastics to interact with various compounds in the environment is appearing with increasing frequency in the literature. Adding to the complexity of MF chemistry is predicting and interpreting sorption of metals, flame retardants, organic pollutants, and other compounds in the environment [63–67]. Additionally, organic molecules sorb to plastics with increasing lipophilicity [68, 69], a property with potentially large biological implications. Once in the environment and following ingestion, additives can leach [55] and any sorbed compounds can desorb [8, 70] during passage through the digestive tract. Under this scenario, effects in addition to those of reproduction may be expected. Teasing apart effects of sorbed contaminants in addition to mechanical damages caused by particles and physiological changes from plastic additives is extremely complex. For this reason, we emphasize the need to include pristine plastic controls in future studies investigating contaminants sorbed in the environment.

### 4.3 MF accumulation

In both the preliminary and formal experiments, PES and PP MFs were evident and quantifiable in gut and feces. We expected MFs to become entangled in gill filaments, particularly in the outgrowths of secondary lamellae unique to medaka [34]. However, in the absence of behavioral changes and MFs in gills during dissection, their passage through the branchial chamber was unclear. The subsequent flushing of MF solution into mouth cavity verified that MFs indeed passed through the branchial chamber and over gills but did not become entwined around them. Localizing MFs in histological sections supported these observations.

In the few laboratory studies of MFs in fish, only one type of MF polymer was studied. We found interesting differences in egestion based on the type of MF. Fish excreted an overall greater number of PES MFs than PP MFs. Amounts of PES egested did not change over time, but while excreted PP MFs overall were less, number did increase over time. This lesser abundance of PP relative to PES MFs might be explained by their density. Density of plastic particles determines location in the water column and affects bioavailability [71]. Although MFs were mixed via air stones, some separated within the water column. Low-density PP floated at the surface and stuck to tank walls while higher-density PES MFs settled on the tank bottoms. It is possible that some MFs may have adhered to feces, increasing measured values. However, preliminary observations did not show MFs in water collected with feces. Medaka have an upturned mouth that allows for feeding at the water's surface [32] and likely ingest floating PP MFs along with their dry diet. When surface food has been exhausted, medaka will search the bottom of tanks for sunken food particles, and this is probably when they ingested additional PES MFs. Normal swimming behavior as well as foraging for *Artemia* nauplii occurs midwater column, where contact with suspended MFs occurred. It was also possible that MF physical characteristics were a critical determinant in ingestion. While similar in length, PP were larger in diameter (50–60 μm) than PES MFs (10–20 μm). Such selectivity in size and/or shape has been reported in goldfish found to chew then expel fragments but to ingest and retain fibers [24].

## 4.4 Gills and branchial chamber

Responses to MFs were most severe along outflow pathways over gills. The morphological alterations we observed are common symptoms of toxic effects in fishes resulting from a variety of aquatic pollutants and are routinely secondary to toxic interaction with specific transport steps or membrane-bound receptors [25, 72].

Typically overlooked are the margins of the branchial chamber. Within the branchial chamber, we observed swollen spaces beneath the inner opercular epithelium, probably arising from interactions with MFs as water followed the inner wall of the operculum before exiting the chamber. Such an effect may disrupt or inhibit osmoregulation by the inner opercular membrane, specifically ion transport and kinetics of its chloride cells [73, 74]. This is the first report of such separation of the inner opercular epithelium from deeper wall structures. Such swellings possibly reduced the volume of the branchial chamber and inhibited water flow. Additionally, we observed this lifting to deform primary and secondary lamellae, likely impairing respiration, and resulting in damage.

Tissue and cellular effects resulting from microplastic exposure have also received very little attention. Results of our SEM and histological investigations showed acute responses including epithelial lifting, increased mucus production, and eroded epithelium as well as chronic responses including erosions on surfaces of gill arches, lamellar aneurysms, and fusion of primary and secondary lamellae [75–77]. Separation of epithelium from the basal membrane is a symptom of disorders of osmoregulation and can act as a protective mechanism to increase distance from toxicants [78], but increased distance also impairs oxygen uptake [72]. Likewise, fusion of lamellae causes an overall reduction in surface area for gas exchange [79]. Increased mucus production also functions as a barrier against foreign substances (chemical, physical, or biological) [75, 76], forming an important part of the innate immune system [76]. While mucus production is considered a defense mechanism, any change that decreases filament surface area or increases distance for gaseous exchange between external environment and blood is regarded as potentially harmful to host respiration [27]. We found rare petechiae in control fish but treated individuals had pronounced and numerous aneurysms. Petechiae that are minor in size and extent, as seen in controls, are reversible changes [72, 80]. Lamellar aneurysms and complete lamellar fusions are severe pathologies [72, 81]. Lamellar aneurysms result in damage and loss of pillar cells in these areas result in the fusion of capillaries within secondary lamellae, which causes their dilation and congestion with blood [75]. Causative factors of gill aneurysms include mechanical injuries or a long list of toxicants that impair respiration [75, 82, 83].

The changes we observed may have been from mechanical damage, responses to leached additives, or a combination of the two. The textile industry employs numerous synthetic dyes (>10,000), some of which are non-biodegradable and carcinogenic [84]. For example, benzothiazoles (BTs), found in many textiles [11], induced gill alterations including epithelial lifting, epithelial hypertrophy, and fusion of secondary lamellae sheepshead minnow (*Cyprinodon variegatus*) larvae [85]. However, there are few studies of this nature that have investigated physical effects of leached additives.

We considered the possibility of recovery from these phenotypic traits should fish be moved to clean water. Recovery of aneurysms is somewhat controversial [81]. Severe changes such as these are often irreversible even when water quality improves [72, 80]. That said, there are some reports of recovery after transfer to clean water. For example, *Hypostomus francisci* (a Brazillian catfish sp.) collected from a polluted river exhibited epithelial hypertrophy and lifting, lamellar fusion, aneurysms, hyperemia, and vascular congestion [81]. While recovery was slow after placement in clean water, full recovery of lamellar aneurysms occurred after 30 days and apoptosis was stimulated to promote gill structure recovery [81]. In a laboratory

study, aneurysms developed on tips of primary lamellae of *Prochilodus scrofa* (a tropical teleost fish) exposed to copper for 96 hrs, with additional damage in the form of epithelial lifting, cell swelling, and proliferation of pavement, chloride, and mucous cells [86]. Again, recovery was slow after transfer to clean water (30–45 days), but much of this damage was reversible [86]. Such recovery studies have not been conducted for microplastics.

### 4.5 Gut

Microfibers are pervasive in digestive tracts of various wild caught fish [21, 24, 87, 88]. In the laboratory, Grigorakis et al. [23] determined retention times for MFs to be fairly low. Our study found MFs were primarily oriented longitudinally within lumina of all gut regions likely favoring rapid passage. Because medaka are agastric teleosts, our examinations were done in three intestinal segments following the description of medaka gut [33]. SEM and AB-PAS stained sections of exposed individuals showed that mucous cells and mucus production increased, primarily in foregut. MFs were encapsulated within luminal mucus and digesta throughout the gut. We hypothesize that this lubricated the gut wall to reduce abrasion and was protective in that it reduced contact with luminal epithelium, facilitating MF passage and excretion [89]. Correspondingly, H&E stained sections showed no significant lesions in intestinal segments of exposed fish.

Interestingly, SEM showed grooves or scratches on the surface of PP fibers in the hindgut lumen. We initially considered that these could be explained on the basis of tooth action during mastication; however, MFs in foregut showed no surficial alterations. We regard contraction of circular and longitudinal muscles of gut wall, as factors increasing contact between MFs and adjacent material of smaller diameter, as the most likely explanation. The formation of such grooves on MFs might release smaller particles from the increased surface area, both of which could lead to enhanced release of fiber additives and subsequent toxicity. We do not believe a significant amount of MF breakage occurred during ingestion and passage based on the finding that MFs recovered from feces did not differ in length from those at initiation of exposure.

## 5 Conclusion

While several field studies report MFs to account for the majority of microplastics both in environmental media and biota, there is a lack of laboratory studies. In adult medaka, we examined multiple levels of biological organization following chronic, aqueous exposure to two types of MFs. Large numbers of MFs were shown to pass through both branchial chamber and gut. Responses in cells and tissues led us to conclude that MFs are potentially harmful to fish and that MF type is an important consideration in toxicity. The branchial chamber, in particular, was the site of both acute and chronic responses. Structural alterations of inner opercular membrane, rakers, and primary- and secondary lamellae were evidence of damage. If presented with other challenges (e.g., predators, hypoxia, competition with other males for spawning), these changes would likely impact survival. Effects observed in other organs (e.g., fecundity) suggest a possible interaction with substances leaching from MFs in gut. Use of a small laboratory model fish has enabled detailed, high resolution investigations of various organs and tissues. We are currently analyzing water and tissue samples generated from this study to answer questions of chemical contributions to toxicity.

## Supporting information

**S1 Fig. Standard curves.** Different concentrations of PP (A) and PES (C) MFs dispersed in 10 mL 70% ethanol. Standard curves of PP (B) and PES (D) MFs.
(DOCX)

**S2 Fig.** Body weights of female (A) and male (B) medaka before (light grey bars) and after exposure (dark grey bars). Medaka were exposed to 0 (Control), PP, or PES MFs for 21 days (n = 18). Data are presented as means ±SD. Mann-Whitney *U*-test and Wilcoxon tests were used to determine the differences in the body weight of medaka among different treatment groups and between before and after exposure, respectively. # p < 0.05, ## p < 0.01.
(DOCX)

**S3 Fig. Embryo survival and hatching.** Survival rate (A-C) and hatching percent (D-F) of embryos collected at day 7 (A, D), 14 (B, E) and 21 (C, F). Data are presented as means, n = 5–9 tanks. PP, Polypropylene MFs; PES, Polyester MFs.
(DOCX)

**S4 Fig.** Malformation rates of larvae at 14 days post fertilization (dpf) from control, PP, and polyester PES MFs for 7 (A), 14 (B) and 21 (C) days. Data are presented as medians, n = 5–9 tanks.
(DOCX)

**S5 Fig. Body lengths of larvae.** Body length at 14 days post fertilization (dpf) larvae exposed to MFs for 14 and 21 days. Data are presented as medians ± SD, n = 5–9 tanks.
(DOCX)

**S6 Fig. Histological micrographs of MF distribution in medaka.** H&E stained sections of mouth (A), buccal cavity (B) and pharynx near teeth (C) from control fish. H&E stained sections of mouth (D), buccal cavity with high magnification inset of MF (E), pharynx near teeth with high magnification inset of MF (F), gill filaments with high magnification inset of MF in direct contact with outgrowths on secondary lamella (G) and gut (H) from PES-exposed fish. AB-PAS stained sections of gut (I) from PP-exposed fish with wall of gut at bottom of field and gut lumen occupying middle to upper portions of field; PP MFs in negatively stained, clear spaces signifying former presence of MFs. Low and high magnification images with black arrows indicate MFs.
(DOCX)

**S7 Fig.** AB-PAS stained histological sections in foregut after 21-day exposure to 0 (control; A-B), PES (C-D), or PP (E-F) MFs. (B, D, F) The higher magnification views of areas in the foregut indicated by squares in A, C, and D. Black arrows indicate goblet cells.
(DOCX)

**S1 Table. Test concentrations.** Mass concentration (mg/L) of PP and PES MFs at the test concentration of 10,000 microfibers/L used for this study.
(DOCX)

**S2 Table. Measurements of adult fish exposed to MFs for 21 days.**
(DOCX)

## Acknowledgments

We would like to thank J. Mac Law at NC State University College of Veterinary Medicine for his consultation on morphological alterations. We would also like to thank Michelle Plue for her consultation on scanning electron microscopy and Mei Zhu for her assistance during the experiment.

## Author Contributions

**Conceptualization:** Lingling Hu, Melissa Chernick, Anna M. Lewis, David E. Hinton.

**Formal analysis:** Lingling Hu.

**Funding acquisition:** Lingling Hu, David E. Hinton.

**Investigation:** Lingling Hu, Melissa Chernick.

**Methodology:** Lingling Hu, Melissa Chernick, Anna M. Lewis, P. Lee Ferguson, David E. Hinton.

**Project administration:** David E. Hinton.

**Resources:** David E. Hinton.

**Supervision:** P. Lee Ferguson, David E. Hinton.

**Visualization:** Melissa Chernick.

**Writing – original draft:** Lingling Hu, Melissa Chernick.

**Writing – review & editing:** Lingling Hu, Melissa Chernick, Anna M. Lewis, P. Lee Ferguson, David E. Hinton.

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
