## [Decision Letter · Decision Letter 0]

19 Dec 2019

PONE-D-19-29596

Chronic microfiber exposure in adult Japanese medaka (*Oryzias latipes*)

PLOS ONE

Dear Dr. Hinton,

Thank you for submitting your manuscript to PLOS ONE. After careful consideration, we feel that it has merit but does not fully meet PLOS ONE’s publication criteria as it currently stands. Therefore, we invite you to submit a revised version of the manuscript that addresses the points raised during the review process.

This is an interesting and well programmed study on a hot topic. I agree with the two reviewers that it is technically robust and I noted that all the data underlying the study are available in a public repository.  However, there is need for a number of clarifications/corrections before the manuscript can be accepted for publication. During the manuscript revision, in addition to the all the comments of the two reviewers, please take into consideration also the following:

Line 197 “…. specimens were immersed in 10x volume of formalin….” What does 10x volume formalin means? Maybe 10%

Line 198 ”fixed at room temperature overnight and then stored at 4ºC until time of processing (section 2.7”). How were the samples sored? Where they removed from formalin?

Line 255 “Mann-Whitney *U*-test was used to determine differences in the quantities of MFs and embryos, *K*, GSI, HSI, body weight and length of medaka among different treatment groups.”

Why did you use a non-parametric test?

Please carefully check the statistics paragraph. It seems that not all the examined parameters have been reported in the statistic paragraph.  

Line 266 “Behavior (*e.g.*, increased opercular movements, erratic swimming, gasping at surface, cowering) was evaluated and no changes were observed.” Please report methods and results for these observations or remove the sentence.

Line 271 Females exposed to PP MFs produced more eggs over the course of the experiment, becoming significantly higher than other treatments by the last week (Fig 2A, p=0.013). How were egg productions compared? This is not reported in the Material and methods section.

Line 302. It seems that only 2 fish were treated in the preparatory experiments. Are the observed anomalies of branchial cavity and gills referred to both the treated fish? What about the number of fish showing all the anomalies reported after 21 days exposure to MF? All the treated fish showed all the reported anomalies? This is not clear. I would suggest adding a table summarizing these findings. Were not gut mucus and globlet cells found in controls?   Why the micrographs regarding these findings are not provided in the manuscript as normal high resolution micrographs? 

Line 398. Rochman, Kurobe (48)…Please uniform this reference to the journal’s style.

We would appreciate receiving your revised manuscript by Feb 02 2020 11:59PM. To enhance the reproducibility of your results, we recommend that if applicable you deposit your laboratory protocols in protocols.io, where a protocol can be assigned its own identifier (DOI) such that it can be cited independently in the future. For instructions see: http://journals.plos.org/plosone/s/submission-guidelines#loc-laboratory-protocols

A rebuttal letter that responds to each point raised by the academic editor and reviewer(s). This letter should be uploaded as separate file and labeled 'Response to Reviewers'.*A marked-up copy of your manuscript that highlights changes made to the original version. This file should be uploaded as separate file and labeled 'Revised Manuscript with Track Changes'*.An unmarked version of your revised paper without tracked changes. This file should be uploaded as separate file and labeled 'Manuscript'.

We look forward to receiving your revised manuscript.

Kind regards,

Aldo Corriero, Ph.D.

Academic Editor

PLOS ONE

Journal Requirements:

1.When submitting your revision, we need you to address these additional requirements.

2. Our internal editors have looked over your manuscript and determined that it may be within the scope of our Plastics in the Environment Call for Papers. The Collection will encompass a diverse range of research articles to better understand various aspects of the effect of plastics in the environment. Additional information can be found on our announcement page: https://collections.plos.org/s/plastics-environment. If you would like your manuscript to be considered for this collection, please let us know in your cover letter and we will ensure that your paper is treated as if you were responding to this call. If you would prefer to remove your manuscript from collection consideration, please specify this in the cover letter."

3. In your Methods section, please provide additional information regarding the processes for euthanasia.

Reviewers' comments:

Reviewer's Responses to Questions

**Comments to the Author**

1. Is the manuscript technically sound, and do the data support the conclusions?

Reviewer #1: Yes

Reviewer #2: Yes

2. Has the statistical analysis been performed appropriately and rigorously? 

Reviewer #1: Yes

Reviewer #2: N/A

3. Have the authors made all data underlying the findings in their manuscript fully available?

Reviewer #1: No

Reviewer #2: Yes

4. Is the manuscript presented in an intelligible fashion and written in standard English?

Reviewer #1: Yes

Reviewer #2: Yes

5. Review Comments to the Author

Reviewer #1: Review:

The topic of this study is worthwhile. There has been little toxicity work on microfibers compared to other types of microplastics. Moreover, there is a need for more chronic toxicity testing in fish with microplastics of any shape/type.

This is also a nice study in that they looked at reproductive endpoints, explored effects in the F1 generation and looked closely at transport and effect using histology.

Overall, this is technically sound based upon some clarification about methods and assumptions being made.

Most people use PET or PEST for polyester. I’d use something more commonly used versus a new acronym. I think technically it’s PET, but I think people maybe steer away from this because of the non-textile PET. Still, they are the same polymer.

How have the authors decided that 10,000/L is environmentally relevant? I don’t think what is in Arctic ice is at all relevant to exposure in fish. Why not just be honest that 1000 is environmentally relevant (on the high end!) and that 10,000 is an extreme (maybe future? concentration).

The introduction reads well.

In methods:

Because it’s more common in the field, I’d provide the average length in micrometers. I’d also suggest giving the standard deviation. I think that is what authors really want to see, in addition to what you have provided.

For the count to mass – does this mean you built a regression off of three points? Can this be described a bit better to understand how many different masses were run in triplicate? If just one, I don’t think a regression is appropriate.

In the SI, I see 5 points, and 5 beakers, (but not three points per mass) so I’m confused what you did. Can you please elaborate on this more in the text?

Prelim study: (can you state why you did a prelim study in first sentence, it’s not clear)

8-month old adults were used. Breeding pairs. Fish were fed two times per day. C, 1000 (PET or PP), 10000 (PET or PP). n=2; exposure time = 21 days.

It’s not well stated how microfibers were added to the tanks. Seems they were simply added to the water and always there. But, when you siphon, how do you decide how much more to add to the tanks?? Are you assuming homogeneity in the water when you remove 25%? Can you say anything about how the actual exposure changed over time? (okay – now I see this is below, but could be described better above so a reader doesn’t get confused – you can describe this was a test to inform how to do additions of microfibers).

When fecal material was removed, how were you sure you just removed fecal material and not fibers that simply settled on the bottom? PET is negatively buoyant and PP is positively buoyant. I’m curious how you can be sure that is what was actually egested versus just in the tank.

This also brings me to a question about how the MFs behaved in the beakers. Did the two types sit in different places in the beaker, thus altering exposure between the two types? I’m just curious, but so may be another reader.

Actual study: 33 breeding pairs, 27 of which were selected based on breeding status. For experiment, used an n=9. Other parameters same as above. They did a hatch out of fertilized eggs and raised them to 14 days post-fertilization. Then measured body length.

In adults, histopath done on 3 breeding pairs per treatment. Was this decided at random?

3 were also taken for SEM.

For other 3, taken for chem analysis.

Please provide how many were used for condition indices? Was this an n=6?

When tank water was sampled for chemical analysis, did you filter out fibers first?

For stats, can you provide your n for each test and can you state why you used non-parametric statistics.

Results:

How was behavior evaluated? Systematically, or just anecdotally you observed no changes?

Again, I really think you should explain how you knew MFs were in feces versus just landed on feces.

Each time a histological change is reported I’d state in how many fish of the fish examined.

The greater amount of PET excreted may be because they sink and were mingling with the feces at the bottom of the tank. It may have nothing to do with bioavailability unless you accounted for this somehow. I suggest discussing this – as mentioned twice above.

If the data is to be fully available, I do not think that it is.

Reviewer #2: This manuscript examined effects of microfiber ingestion on medaka. The experiment was designed well. A thorough assessment was made of MF entry, egress, and interaction with tissues as they passed through head gut, branchial chamber, and digestive system. Ultra-structural changes in the gill and gut were examined. Findings suggest that MF ingestion does not affect major physiological processes including reproduction but induce aneurysms in secondary lamellae, epithelial lifting, and swellings of inner opercular membrane that altered morphology of rostral most gill lamellae. Increased numbers of mucous cells and secretions on epithelium of foregut was observed but without overt abrasions with sloughing of cells. Results suggest that microfiber ingestion causes adverse effects on gills and the gut in fish.

I suggest authors to address a few issues. The findings that PP MF ingestion caused increased fecundity and fertilization success in fish sounds interesting. The mechanism underlying this change has been specuclated to be estrogenicity of plastic MFs. This speculation should be removed from the abstract.

Line 165: “most consistent productivity were selected”. Please mention what these parameters were.

Line 180: “Eggs were collected by siphoning 24 h after complete water changes and examined to determine whether MFs in tank water had become incorporated in egg clutches.” Medaka tend to eat their eggs after spawning. If eggs were siphoned from the bottom, then the numbers could be inaccurate. To avoid this, eggs should be collected directly from the fish 30 minutes after spawning. Also, spawning time is not mentioned. Was the spawning sychronized? Usually medaka lay eggs within an hour of lights on in the morning.

Line 333-339: “Petechiae (i.e., small spots of hemorrhage) and epithelial lifting were found in gills of 50% of control fish, but were minor in size and extent, with rare petechiae in different positions along the gill filament. Conversely, aneurysms and epithelial lifting occurred in gills of 67% of PES-treated and 83% of PP-treated fish and were numerous and mainly concentrated along water outflow tracts (i.e., passages between adjacent gill arches and their associated primary lamellae) (Fig 4E). Fusion of secondary lamellae was observed in MF-treated fish, most frequently (67%) after PP exposure (Figs 4H and 5H)”. Would you expect a recovery of all these phenotypic traits in these fish if transferred to water for 21 days without MFs? Addition of this piece of data would strengthen the finding of this manuscript.

Line 393-413: Fecundity part of discussion is extremely speculated. Have you found the leached amount of BPA or phthalates from MFs to be effective enough to induce these physiological changes in 21 days of exposure? This part of discussion needs to be rewritten minimize speculation if supporting evidence is not available.

How about the possibility for loading of other chemical contaminants into the body together with microfibers ingestion in a natural situation? Addition of this information would strengthen the quality of this manuscript.

6. PLOS authors have the option to publish the peer review history of their article (what does this mean?). If published, this will include your full peer review and any attached files.

Reviewer #1: No

Reviewer #2: No

---

## [Author Response · Author response to Decision Letter 0]

24 Jan 2020

Editor’s Comments

This is an interesting and well programmed study on a hot topic. I agree with the two reviewers that it is technically robust and I noted that all the data underlying the study are available in a public repository. However, there is need for a number of clarifications/corrections before the manuscript can be accepted for publication. During the manuscript revision, in addition to the all the comments of the two reviewers, please take into consideration also the following.

Response: We thank the editor for the positive feedback. We have addressed editor’s concerns as well as those of the reviewers in the responses below. We have provided line numbers where appropriate for specific changes; line numbers in Revised Manuscript with Track Changes are preceded by “TC” and those in the clean, revised Manuscript are preceded with “M”. We are keeping the original figure uploads with our revision only for peer review. As instructed below, we have uploaded high resolution versions of these figures to PACE.

Specific comments

Line 197 “…. specimens were immersed in 10x volume of formalin….” What does 10x volume formalin means? Maybe 10%

Response: Thank you for the comment. This was meant to express the volume of formalin was tenfold that of specimens. This volume is used to ensure that the 10% concentration is not diluted by fluid transfer from the intact fish, that adequate fixation occurs, and that specimens can be stored long term until processing [1-2]. We have revised it (TC214; M207).

1. Meyers TR. Fish pathology section laboratory manual. 3rd ed. Juneau, AK: Alaska Department of Fish and Game, Commercial Fisheries Division; 2009.

2. Morrison J, Smith C, Heidel J, Mumford S, Blazer VS, MacConnell E. Fish Histology and Histopathology Manual. In: USFWS, editor. Shepardstown, WV: National Conservation Training Center; 2014. p. 357.

Line 198 ”fixed at room temperature overnight and then stored at 4ºC until time of processing (section 2.7”). How were the samples stored? Where they removed from formalin?

Response: The specimens were placed in 50-mL centrifuge tubes (1 breeding pair/tube) filled with 10% neutral buffered formalin. These tubes remained at room temperature overnight for the formalin to penetrate/fix the specimens. Then the tubes were moved to a refrigerator (4ºC) to be stored until processing for histology (TC214-217; M207-209). Specimens were not removed from formalin. We have edited this sentence to increase clarity. 

Line 255 “Mann-Whitney U-test was used to determine differences in the quantities of MFs and embryos, K, GSI, HSI, body weight and length of medaka among different treatment groups.”

Why did you use a non-parametric test?

Please carefully check the statistics paragraph. It seems that not all the examined parameters have been reported in the statistic paragraph. 

Response: Thank you for the comment. Kolmogorov-Smirnov and Shapiro-Wilk tests were performed to test for normality, and a Levene test was used for homogeneity of variance. Our data were not normally distributed and had unequal variance. Therefore, we used non-parametric tests. We have added this detail to section 2.9 (TC273-285; M265-275). We have verified the tests listed in section 2.9 and edited it to make sure all presented data are listed.

Line 266 “Behavior (e.g., increased opercular movements, erratic swimming, gasping at surface, cowering) was evaluated and no changes were observed.” Please report methods and results for these observations or remove the sentence.

Response: During our experiment, we employed our normal assessment that we use for our colony, which includes daily observations for these signs of stress. We did not observe alterations in these daily assessments. Thus, we have moved this sentence to the Methods section 2.5 (TC147-149; M145-147) and revised it.

Line 271 Females exposed to PP MFs produced more eggs over the course of the experiment, becoming significantly higher than other treatments by the last week (Fig 2A, p=0.013). How were egg productions compared? This is not reported in the Material and methods section.

Response: We wanted to make sure that we began our experiment with females that produced the same number of eggs and males that were able to successfully fertilize those eggs. Reproduction also needed to be high, both in egg numbers as well as fertilization success, and consistent so that we did not mistake variability with a microfiber related effect. These numbers also served as baseline “Before Exposure” values. In this way, we were able to compare these endpoints from different exposure weeks with that baseline within the same treatment group (Fig. 2). These methods about embryo collection and assessment are sections 2.4 (TC174-175; M171-172) and 2.5 (TC195-197; M190-194). Production was compared both between treatment groups (effect of exposure/MF type) as well as within treatment groups (effect of exposure time) (TC277-280; M269-272). The legend for Fig. 2 provides additional information regarding this analysis. We have added details to these areas that should increase clarity. 

Line 302. It seems that only 2 fish were treated in the preparatory experiments. Are the observed anomalies of branchial cavity and gills referred to both the treated fish? What about the number of fish showing all the anomalies reported after 21 days exposure to MF? All the treated fish showed all the reported anomalies? This is not clear. I would suggest adding a table summarizing these findings. Were not gut mucus and globlet cells found in controls? Why the micrographs regarding these findings are not provided in the manuscript as normal high resolution micrographs? 

Response: A preliminary study was conducted in order to determine 1) whether aqueous exposures to MFs would result in uptake and 2) how and in what quantities MFs should be used (a sentence now added to the beginning of section 2.3: TC127-128; M126-127). For this preliminary, 2 breeding pairs (4 fish) were in each treatment group. SEM and histology were not performed on preliminary fish. During dissections, we looked carefully for MFs inside the fish under a stereomicroscope. Digestion of excised organs using H2O2 was performed (TC160-161, 166-171; M157-158,169-168) to determine presence of MFs. Females were fixed in formalin and stored as described in section 2.6 in case we needed to go back and re-evaluate. It turned out that this was not needed. Instead of another table in Supplemental Information, we have added worksheets to our data file publicly available on Figshare to supply preliminary data, additional pictures and figures. 

In the formal experiment, occurrences were high for some changes. For example, 83% of PP exposed fish and 67% of PES exposed fish had epithelial lifting on gills. We report percent occurrences in the text of section 3.4. As such, we consider a table to be redundant, but should the editor still deem it necessary we are happy to add one. 

The editor is correct in that the surface of the gut lumen has a mucus layer secreted by goblet cells within the underlying mucosa [1-2]. As in gills, increased mucus production in gut is considered a first defense strategy to foreign particles [3] and can be visualized when special stains such as AB-PAS are used. The increase in size and number of goblet cells indicates a response to MFs. Such increases result in additional mucus to provide a physical barrier, prevent mechanical damage, and facilitate passage of MFs through the gut [4-5]. We believe this mechanism was successful in protecting gut tissues from damage, which was not the case in branchial chamber.

Considering the types and degree of changes we observed, we consider the branchial chamber to be the primary area affected. As we already have 7 multi-paneled figures, one of which is SEM of gut, we did not feel it necessary to add the micrographs in S7 Fig to the main manuscript. However, if the editor believes the manuscript is better served with it there, we can move it out of Supporting Information.

Regarding micrographs in general: We are uploading high resolution image files for all figures to the PACE digital tool so that readers may zoom in to insets and other areas of interest without pixelating the image. 

1. Linden SK, Sutton P, Karlsson NG, Korolik V, McGuckin MA. Mucins in the mucosal barrier to infection. Mucosal Immunology. 2008;1:183. doi: 10.1038/mi.2008.5.

2. Sundh H, Sundell KS. Environmental impacts on fish mucosa. In: Beck BH, Peatman E, editors. Mucosal Health in Aquaculture. San Diego, CA: Academic Press; 2015. p. 171-97. doi: 10.1016/B978-0-12-417186-2.00007-8

3. Pedà C, Caccamo L, Fossi MC, Gai F, Andaloro F, Genovese L, et al. Intestinal alterations in European sea bass Dicentrarchus labrax (Linnaeus, 1758) exposed to microplastics: Preliminary results. Environmental Pollution. 2016;212(Supplement C):251-6. doi: 10.1016/j.envpol.2016.01.083.

4. Peterson TS. Overview of mucosal structure and function in teleost fishes. In: Beck BH, Peatman E, editors. Mucosal Health in Aquaculture. San Diego, CA: Academic Press; 2015. p. 55-65. doi: 10.1016/B978-0-12-417186-2.00003-0

5. Shephard KL. Functions for fish mucus. Rev Fish Biol Fisheries. 1994;4(4):401-29.

Line 398. Rochman, Kurobe (48)…Please uniform this reference to the journal’s style.

Response: It appears that the downloaded EndNote citation template has this error. We have corrected the in-text citations to the journal’s style.

Journal Requirements:

Response: We have thoroughly checked the manuscript to ensure it meets these requirements.

2. Our internal editors have looked over your manuscript and determined that it may be within the scope of our Plastics in the Environment Call for Papers. The Collection will encompass a diverse range of research articles to better understand various aspects of the effect of plastics in the environment. Additional information can be found on our announcement page: https://collections.plos.org/s/plastics-environment. If you would like your manuscript to be considered for this collection, please let us know in your cover letter and we will ensure that your paper is treated as if you were responding to this call. If you would prefer to remove your manuscript from collection consideration, please specify this in the cover letter."

Response: We agree that our study is a good fit with the Plastics in the Environment Call for Papers. We would like for it to be considered for this collection. We have also indicated this in our cover letter.

3. In your Methods section, please provide additional information regarding the processes for euthanasia.

Response: We have added additional description and the citations below for the rapid cooling method for euthanasia at its first mention in Methods section 2.3 (TC167-168; M163-165). Rapid cooling (or hypothermal shock) is considered to be a less stressful and more rapid method of euthanasia than MS-222 [1]. It is recommended by the American Veterinary Medical Association (AVMA) for finfish [2, see its section S2.5] and was approved by the Duke University Institutional Animal Care and Use Committee (IACUC) for this study.

1. Matthews M, Varga ZM. Anesthesia and Euthanasia in Zebrafish. ILAR J. 2012;53(2):192-204. doi: 10.1093/ilar.53.2.192.

2. Leary SL, Underwood W, Anthony R, Cartner S, Corey D, Grandin T, et al., editors. AVMA Guidelines for the Euthanasia of Animals: 2013 Edition. 2013: American Veterinary Medical Association, Schaumburg, IL. https://www.avma.org/sites/default/files/resources/euthanasia.pdf

Reviewer #1:

The topic of this study is worthwhile. There has been little toxicity work on microfibers compared to other types of microplastics. Moreover, there is a need for more chronic toxicity testing in fish with microplastics of any shape/type.

This is also a nice study in that they looked at reproductive endpoints, explored effects in the F1 generation and looked closely at transport and effect using histology.

Overall, this is technically sound based upon some clarification about methods and assumptions being made.

Response: We thank the reviewer for the positive feedback and specific comments, we have addressed each in the responses below. We have provided line numbers where appropriate for specific changes; line numbers in Revised Manuscript with Track Changes are preceded by “TC” and those in clean, revised Manuscript are preceded with “M”.

Most people use PET or PEST for polyester. I’d use something more commonly used versus a new acronym. I think technically it’s PET, but I think people maybe steer away from this because of the non-textile PET. Still, they are the same polymer.

Response: We agree with the reviewer that PET is used as an acronym for polyester. However, polyester is a category of polymer that includes polyethylene terephthalate, also commonly abbreviated as PET. We have seen the abbreviation PES to refer to polyester (fibers in particular) in several journal articles [e.g., 1-4], much fewer PEST [5]. Therefore, to avoid confusion with other polymers, we use PES to distinguish it.

1. de Sá LC, Oliveira M, Ribeiro F, Rocha TL, Futter MN. Studies of the effects of microplastics on aquatic organisms: What do we know and where should we focus our efforts in the future? Science of The Total Environment. 2018;645:1029-39. doi: 10.1016/j.scitotenv.2018.07.207.

2. Duis K, Coors A. Microplastics in the aquatic and terrestrial environment: sources (with a specific focus on personal care products), fate and effects. Environmental Sciences Europe. 2016;28(1):2. doi: 10.1186/s12302-015-0069-y.

3. Ivleva NP, Wiesheu AC, Niessner R. Microplastic in aquatic ecosystems. Angewandte Chemie International Edition. 2017;56(7):1720-39. doi: 10.1002/anie.201606957.

4. Deng H, Wei R, Luo W, Hu L, Li B, Di Yn, et al. Microplastic pollution in water and sediment in a textile industrial area. Environmental Pollution. 2020;258:113658. doi: 10.1016/j.envpol.2019.113658.

5. Rummel CD, Löder MGJ, Fricke NF, Lang T, Griebeler E-M, Janke M, et al. Plastic ingestion by pelagic and demersal fish from the North Sea and Baltic Sea. Marine Pollution Bulletin. 2016;102(1):134-41. doi: 10.1016/j.marpolbul.2015.11.043.

How have the authors decided that 10,000/L is environmentally relevant? I don’t think what is in Arctic ice is at all relevant to exposure in fish. Why not just be honest that 1000 is environmentally relevant (on the high end!) and that 10,000 is an extreme (maybe future? concentration).

Response: We agree with the reviewer that 10,000/L is a high concentration of MFs. We used “environmentally relevant” in this study to refer to concentrations detected in the environment (i.e., Arctic ice). We also agree that “extreme or future concentration” may be a better way to refer to it. To avoid misunderstanding, we have deleted “environmentally relevant” in the manuscript (TC27-28, 139) and added future concentration to this portion of the Methods (TC139-141; M138-139). 

The introduction reads well.

Response: We thank the reviewer for this comment.

In methods:

Because it’s more common in the field, I’d provide the average length in micrometers. I’d also suggest giving the standard deviation. I think that is what authors really want to see, in addition to what you have provided.

Response: We have changed the units to µm as the reviewer requested. However, we left the units as mm in Fig. 1A4-B4 because the additional text required for µm would make the axis labels difficult to read. Note there was the percentage of size distributions of MFs (Fig. 1A4-B4) and no standard deviation.

For the count to mass – does this mean you built a regression off of three points? Can this be described a bit better to understand how many different masses were run in triplicate? If just one, I don’t think a regression is appropriate.

In the SI, I see 5 points, and 5 beakers, (but not three points per mass) so I’m confused what you did. Can you please elaborate on this more in the text?

Response: We set up five gradient masses (5 points on each graph) (TC112; M111). For each mass/point, the filter, imaging and counting processes were repeated in triplicate (TC118-119; M117-118). This gave us three measurements per mass/point. We used the average number per mass to build the regression (S1 Fig B,D). We have added this information to the methods (TC108-120; 107-119) as well as the counts in the data file available on Figshare. See the response below regarding beakers for additional information.

Prelim study: (can you state why you did a prelim study in first sentence, it’s not clear)

8-month old adults were used. Breeding pairs. Fish were fed two times per day. C, 1000 (PET or PP), 10000 (PET or PP). n=2; exposure time = 21 days.

It’s not well stated how microfibers were added to the tanks. Seems they were simply added to the water and always there. But, when you siphon, how do you decide how much more to add to the tanks?? Are you assuming homogeneity in the water when you remove 25%? Can you say anything about how the actual exposure changed over time? (okay – now I see this is below, but could be described better above so a reader doesn’t get confused – you can describe this was a test to inform how to do additions of microfibers).

Response: We thank the reviewer for pointing this out. We have added a sentence to the beginning of section 2.3 (TC127-128; M126-127). 

The reviewer is correct that the MFs were added (dry) to tanks. The amount added was based on volume of water removed and amount of MFs bound to feces (also removed). There was some assumption of homogeneity of MFs in water that was removed during cleaning. Please see the responses below regarding MF addition, quantities in feces, and buoyancy.

When fecal material was removed, how were you sure you just removed fecal material and not fibers that simply settled on the bottom? PET is negatively buoyant and PP is positively buoyant. I’m curious how you can be sure that is what was actually egested versus just in the tank.

Response: We agree that buoyancy was a potentially complicating factor. We considered this when designing and executing our preliminary study. We collected feces with a 7.5-mL transfer pipette to avoid removing too much water volume that might affect MF concentration or skew egestion numbers. Additionally, when we observed our collected samples under a stereomicroscope, we noted that excreted MFs were bound within the feces (Fig. 3) and very few (i.e., negligible number) MFs were in the water collected with them. This was the case with both fiber types. We have added these details to section 2.3 (TC156-158; M153-155).

This also brings me to a question about how the MFs behaved in the beakers. Did the two types sit in different places in the beaker, thus altering exposure between the two types? I’m just curious, but so may be another reader.

Response: This is another good observation by the reviewer and one we also considered in the design and execution of our study. The beakers pictured in S1 Fig A,C show MFs in 70% ethanol, which disperses them evenly within the solution and are just visualizations for the accompanying standard curves (S1 Fig. B,D). During our early characterizations of the MFs, we observed that many MFs suspended in water remained on walls of beakers when water was poured out. The only way to remove all adhered MFs was to add clean water and repour, often multiple times, thereby changing our final MF concentration. We realized that we needed a way to introduce a known number of MFs to tanks based on the amount of water removed during routine tank maintenance as well as those bound to feces (also removed from tanks). The method also needed to be practical because counting 10,000 fibers/L for 9 tanks/treatment for 21 days was time consuming and unrealistic. This was the impetus for the creation of the standard curves shown in S1 Fig, which allowed us to add dry MFs by weight. These curves provided a weight (mg) MFs needed based amount of water removed, and the preliminary study determined the number of MFs bound to removed feces. Additionally, full cleanings every 7 days “reset” tanks in case the number had drifted in one direction or another.

As density of MFs determines location in the water column, it affects bioavailability. Even with mixing provided by air stones, some MFs sank or floated. This was why we felt the need to address it in terms of exposure in the Discussion (TC486-503; M468-485).

Actual study: 33 breeding pairs, 27 of which were selected based on breeding status. For experiment, used an n=9. Other parameters same as above. They did a hatch out of fertilized eggs and raised them to 14 days post-fertilization. Then measured body length.

In adults, histopath done on 3 breeding pairs per treatment. Was this decided at random?

3 were also taken for SEM.

For other 3, taken for chem analysis.

Please provide how many were used for condition indices? Was this an n=6?

Response: The reviewer is correct. We moved 66 fish (33 males and 33 females) from the same age cohort in our breeding colony to our exposure room to make 33 breeding pairs. Under clean conditions, we assessed each pair for reproduction. We started with more pairs than we would ultimately use because we knew a small proportion may not be desirable for the experiment. Any fish that did not reproduce were returned to the colony. Of the confirmed breeders, the 27 pairs that produced consistent, high numbers of fertilized eggs were selected for the study. There were no significant differences in the egg numbers or fertilization rates between treatment groups before exposure. This also provided a baseline for selection and later comparisons over the course of the experiment.

In terms of “randomness,” this was determined at the beginning rather than at the end of the experiment. The 27 pairs that were chosen were assigned randomly to the 3 treatment groups, with 9 pairs (18 fish) per group. Tanks in each group were numbered 1-9. Before the exposure began, tanks 1-3 were allocated to histology, 4-6 to chemistry, and 7-9 to SEM. The type of analysis for each pair/fish was determined based tank number and not on data or observations made during the course of the experiment.

All fish in each treatment group were assessed for condition factor (K). Because organs were not removed from fish allocated to histology, GSI and HSI were from 6 pairs (n=12) from each group. We have added “n=” values where needed in the manuscript (e.g., TC282-283; M273).

When tank water was sampled for chemical analysis, did you filter out fibers first?

Response: We avoided the collection of fibers when sampling tank water. Moreover, the water was filtered (0.2 µm) before instrument detection (TC186-187; M182-184). 

For stats, can you provide your n for each test and can you state why you used non-parametric statistics.

Response: Thank you for the comment. We have added “n=” values where needed. Kolmogorov-Smirnov and Shapiro-Wilk tests were performed to test for normality, and a Levene test was used for homogeneity of variance. Our data were not normally distributed and had unequal variance. Therefore, we used non-parametric tests. We have added this to section 2.9 (TC274-284; M266-274). 

Results:

How was behavior evaluated? Systematically, or just anecdotally you observed no changes?

Response: This was evidently a confusing point as it was brought up by another reviewer. For behavior, we employed our normal assessment that we use for our colony, which includes daily observations for signs of stress. We used these routine observations during the experiment. We did not observe alterations in behavior. As this is not a behavioral study, making this routine observation more of a method, we have moved this sentence to the Methods section 2.5 (TC147-149; M145-147) and revised it.

Again, I really think you should explain how you knew MFs were in feces versus just landed on feces.

Response: We understand the importance of this distinction. We did not observe preferential binding of suspended or sunken MFs to feces (we have added this to section 2.3: TC156-158; M154-155). Observations of feces using a stereomicroscope showed MFs to be bound within fecal material (Fig. 3). Additionally, SEM of gut consistently showed MFs to be encased within digesta (Fig. 6E). As stated above, we collected as little water as possible with the feces and there were negligible numbers of free MFs (TC156-157; M153-154). These would have been MFs that had landed on or very near feces in the tank should that have happened. Therefore, we are confident the MFs we counted from fecal material were those bound within it.

Each time a histological change is reported I’d state in how many fish of the fish examined.

Response: For each treatment group, there were 6 fish (3 males, 3 females) sectioned for histology. We expressed occurrence results as percentages of these 6 fish. We believe that our addition of “n=” values in various portions of the manuscript will clarify this.

The greater amount of PET excreted may be because they sink and were mingling with the feces at the bottom of the tank. It may have nothing to do with bioavailability unless you accounted for this somehow. I suggest discussing this – as mentioned twice above.

Response: We have added sentences in both the Methods and Discussion (TC493-495; M475-477) that address this issue. Our observations of negligible numbers of MFs collected with feces support our conclusion that measured MFs from feces were indeed those bound within it. For this reason, we consider buoyancy of MFs to be a larger factor for ingestion as described in the Discussion (M477-485).

If the data is to be fully available, I do not think that it is.

Response: Our raw data set file is available from the Figshare database (https://figshare.com/s/08259bef8fbc6df4dc2e) with doi: 10.6084/m9.figshare.10031471. All other data and figures are in the manuscript and supporting information.

Reviewer #2: 

This manuscript examined effects of microfiber ingestion on medaka. The experiment was designed well. A thorough assessment was made of MF entry, egress, and interaction with tissues as they passed through head gut, branchial chamber, and digestive system. Ultra-structural changes in the gill and gut were examined. Findings suggest that MF ingestion does not affect major physiological processes including reproduction but induce aneurysms in secondary lamellae, epithelial lifting, and swellings of inner opercular membrane that altered morphology of rostral most gill lamellae. Increased numbers of mucous cells and secretions on epithelium of foregut was observed but without overt abrasions with sloughing of cells. Results suggest that microfiber ingestion causes adverse effects on gills and the gut in fish.

Response: Thank you for the constructive and detailed comments. We have addressed the reviewer’s concerns in the responses below. We have provided line numbers where appropriate for specific changes; line numbers in Revised Manuscript with Track Changes are preceded by “TC” and those in clean, revised Manuscript are preceded with “M”.

I suggest authors to address a few issues. The findings that PP MF ingestion caused increased fecundity and fertilization success in fish sounds interesting. The mechanism underlying this change has been specuclated to be estrogenicity of plastic MFs. This speculation should be removed from the abstract.

Response: Thank you for your suggestion, we have removed this from the abstract (TC32). We have addressed it with additional detail in the Discussion (section 4.2: TC425-440,449-463; M414-427,436-445) and in a response below.

Line 165: “most consistent productivity were selected”. Please mention what these parameters were.

Response: Consistent productivity refers to a female producing the same number of eggs each day and males consistently fertilizing the same percentage of those eggs. We did not want variability in production to influence later measurements and comparisons. We have added the definition for this in section 2.3 (TC176-177; M173-174).

Line 180: “Eggs were collected by siphoning 24 h after complete water changes and examined to determine whether MFs in tank water had become incorporated in egg clutches.” Medaka tend to eat their eggs after spawning. If eggs were siphoned from the bottom, then the numbers could be inaccurate. To avoid this, eggs should be collected directly from the fish 30 minutes after spawning. Also, spawning time is not mentioned. Was the spawning sychronized? Usually medaka lay eggs within an hour of lights on in the morning.

Response: We appreciate the reviewer’s knowledge of medaka. We collected eggs within a 24 h period after complete water changes rather than after 24 h (we have revised this sentence: TC195; M190). We did not want to add handling stress to our fish by removing eggs manually from females. Also, any net used to capture fish would have also captured MFs, reducing concentration. We have maintained this medaka colony for >20 years and have data on spawning times, egg numbers, fertilization rates, deformity rates, and hatching times/success. Most females in our breeding colony strip eggs within ~1 h after lights turn on in the morning, typically ~30 min after feeding. A small proportion of females strip eggs ~30 min after midday feeding (~1-2pm). Observations and counts of experimental fish during the reproductive evaluation both in the preliminary study and before the formal experiment showed this same pattern. The reviewer is correct that medaka tend to eat their eggs after spawning. For this reason, we checked for oviposition before and after feedings, and every 2-3 h, and eggs were collected as soon as they were observed (we have added this information to Methods). Because the proximity of our offices and main laboratory to our exposure room (~25 m), the frequency of these observations was facilitated. The total number of eggs per day, both before the exposure and during, is consistent with the number of eggs medaka females should produce [1]. Finally, we observed no eggs in gut in SEM or histologic sections.

1. Kinoshita M, Murata K, Naruse K, Tanaka M. Medaka: Biology, management, and experimental protocols. Singapore: John Wiley & Sons, Ltd.; 2009.

Line 333-339: “Petechiae (i.e., small spots of hemorrhage) and epithelial lifting were found in gills of 50% of control fish, but were minor in size and extent, with rare petechiae in different positions along the gill filament. Conversely, aneurysms and epithelial lifting occurred in gills of 67% of PES-treated and 83% of PP-treated fish and were numerous and mainly concentrated along water outflow tracts (i.e., passages between adjacent gill arches and their associated primary lamellae) (Fig 4E). Fusion of secondary lamellae was observed in M recovery F-treated fish, most frequently (67%) after PP exposure (Figs 4H and 5H)”. Would you expect a of all these phenotypic traits in these fish if transferred to water for 21 days without MFs? Addition of this piece of data would strengthen the finding of this manuscript.

Response: We thank the reviewer for this observation and agree. Petechiae that are minor in size and extent, as seen in controls are reversible changes [1, 2]. Moderate changes leading to effects associated with organ function can be repaired unless wide areas are affected [1, 2]. Lamellar aneurysms and complete lamellar fusions are severe pathologies [2, 3]. The former results in damage including loss of pillar cells and destruction of lamellae [2-4]. Recovery of aneurysms is somewhat controversial [3]. Severe changes such as these are often irreversible even when water quality improves [1, 2]. That said, there are some reports of recovery after transfer to clean water. For example, after Hypostomus francisci (a Brazillian catfish sp.) from a polluted river exhibited epithelial hypertrophy and lifting, lamellar fusion, aneurysms, hyperemia, and vascular congestion [3]. While recovery was slow after placement in clean water, full recovery of lamellar aneurysms occurred after 30 days and apoptosis was stimulated to promote gill structure recovery [3]. In a laboratory study, aneurysms developed on tips of primary lamellae of Prochilodus scrofa (a tropical teleost fish) exposed to copper for 96 h, with additional damage in the form of epithelial lifting, cell swelling, and proliferation of pavement, chloride, and mucous cells [5]. Again, recovery was slow after transfer to clean water (30-45 days), but much of this damage was reversible [5].

We have added the above to the Discussion (TC543-555; M525-537).

1. Nascimento AA, Araújo FG, Gomes ID, Mendes RMM, Sales A. Fish gills alterations as potential biomarkers of environmental quality in a eutrophized tropical river in south-eastern Brazil. Anatomia, Histologia, Embryologia. 2012;41(3):209-16. doi: 10.1111/j.1439-0264.2011.01125.x.

2. Flores-Lopes F, Thomaz AT. Histopathologic alterations observed in fish gills as a tool in environmental monitoring. Brazilian Journal of Biology. 2011;71(1):179-88. doi: 10.1590/S1519-69842011000100026.

3. Sales CF, Santos KPEd, Rizzo E, Ribeiro RIMdA, Santos HBd, Thomé RG. Proliferation, survival and cell death in fish gills remodeling: From injury to recovery. Fish & Shellfish Immunology. 2017;68:10-8. doi: 10.1016/j.fsi.2017.07.001.

4. Strzyzewska E, Szarek J, Babinska I. Morphologic evaluation of the gills as a tool in the diagnostics of pathological conditions in fish and pollution in the aquatic environment: a review. Veterinární Medicína. 2016;61(3):123-32. doi: 10.17221/8763-VETMED.

5. Cerqueira CCC, Fernandes MN. Gill tissue recovery after copper exposure and blood parameter responses in the tropical fish Prochilodus scrofa. Ecotoxicology and Environmental Safety. 2002;52(2):83-91. doi: 10.1006/eesa.2002.2164.

Line 393-413: Fecundity part of discussion is extremely speculated. Have you found the leached amount of BPA or phthalates from MFs to be effective enough to induce these physiological changes in 21 days of exposure? This part of discussion needs to be rewritten minimize speculation if supporting evidence is not available.

Response: We agree with the reviewer that there is a certain amount of speculation in the linkage of fecundity to leached additives. Currently, our study shows a common biomarker of endocrine disruption in partial life cycle tests: changes in egg number [1-3]. Considering this was not observed in control fish and the MFs did not leave the digestive tract, we are assured that the effects we observed are related to chemicals present within the MFs. The highest amount of speculation is not in the presence/absence of chemicals in MFs, but rather which chemicals were present and leached. As such, we present plausible mechanistic interpretations based on available literature of release rates and laboratory exposures using single chemicals (of which BPA is the most studied) or simple mixtures.

A hazardous substance that remains within plastic has a lower risk; it needs to be leached/released/emitted for toxicity to occur [4]. This can occur in all phases of a plastic’s life cycle, in a variety of media, and can depend on the composition of non-polymeric substances [4]. However, determination of type and magnitude of leaching is complex as it depends on a multitude of factors [4]. Several substances have been studied for release including phthalates and bisphenols. Studies of single plasticizers have reported biological effects at ng/L or µg/L concentrations [5]. In the gastrointestinal tract, release rates are high, especially for species with longer gut retention times, such as fish [6]. Laboratory studies have shown fish can retain microplastics in their digestive tracts anywhere from 3 to 14 days after cessation of exposure [7-9]. Even low doses of these chemicals can disrupt endocrine systems [10], and their presence as mixtures present complications and shown to fit with concentration addition expectations for endocrine disruptors in fish [11]. 

It should be noted that most studies have been done with pristine microspheres or fragments. There is a lack of data about the actual content of additives in textiles in the common market, primarily due to difficulties in obtaining information from producers on substances used during manufacturing [12]. For example, benzotriazoles (BTris), abundant in clothing textiles, are persistent in the environment and are known to have bioaccumulative properties [12]. Following aqueous exposure to BTris (0.01-1 mg/L) for 4 or 35 days, adult marine medaka (Oryzias melastigma) had induced vitellogenin (VTG) gene expression in liver, gills, and gut of both sexes, down-regulated CYP1A1 gene expression levels in liver and gut, and induced CYP19a expression in ovaries [13]. Those results indicate BTris is an endocrine disruptor in that VTG production is estrogen dependent, many estrogenic chemicals have been reported to inhibit CYP1A1, and CYP19a is involved the control of various physiological functions of estrogens [13]. Importantly, the exposure duration of 35 d, similar to our 21 d, showed responses that a shorter exposure (4 d) did not. This lends support to our contention.

Knowing the wide range of chemicals that were likely present in the MFs used in this study, we purposefully designed the experiment so that tissues could be analyzed for chemicals that would be used to explain the responses we observed. Currently, we are identifying compounds present in the MFs used in this study and their leaching rates so that they can be compared to these collected tissues. Due the complexity of the chemical analyses and the large volume of data it is producing, we have decided to publish it as a follow-up to this study. We have integrated several of the above sentences into the Discussion (TC425-440,449-463; M414-427,436-445) to clarify this issue.

1. Ankley GT, Johnson RD. Small fish models for identifying and assessing the effects of endocrine-disrupting chemicals. Ilar J. 2004;45(4):469-83. PubMed PMID: WOS:000224480300010.

2. Denslow N, Sepúlveda M. Ecotoxicological effects of endocrine disrupting compounds on fish reproduction. In: Babin PJ, Cerdà J, Lubzens E, editors. The Fish Oocyte: From Basic Studies to Biotechnological Applications. Dordrecht: Springer Netherlands; 2007. p. 255-322.

3. OECD. Detailed review paper on fish screening assays for the detection of endocrine active substances. Paris, France: Organisation for Economic Cooperation and Development; 2004.

4. Hahladakis JN, Velis CA, Weber R, Iacovidou E, Purnell P. An overview of chemical additives present in plastics: Migration, release, fate and environmental impact during their use, disposal and recycling. Journal of Hazardous Materials. 2018;344:179-99. doi: 10.1016/j.jhazmat.2017.10.014.

5. Oehlmann J, Schulte-Oehlmann U, Kloas W, Jagnytsch O, Lutz I, Kusk KO, et al. A critical analysis of the biological impacts of plasticizers on wildlife. Philosophical Transactions of the Royal Society B: Biological Sciences. 2009;364(1526):2047-62. doi: 10.1098/rstb.2008.0242. PubMed PMID: PMC2873012.

6. Koelmans AA, Besseling E, Foekema EM. Leaching of plastic additives to marine organisms. Environmental Pollution. 2014;187:49-54. doi: 10.1016/j.envpol.2013.12.013.

7. Ory NC, Gallardo C, Lenz M, Thiel M. Capture, swallowing, and egestion of microplastics by a planktivorous juvenile fish. Environmental Pollution. 2018;240:566-73. doi: 10.1016/j.envpol.2018.04.093.

8. Cong Y, Jin F, Tian M, Wang J, Shi H, Wang Y, et al. Ingestion, egestion and post-exposure effects of polystyrene microspheres on marine medaka (Oryzias melastigma). Chemosphere. 2019;228:93-100. doi: 10.1016/j.chemosphere.2019.04.098.

9. Zhu M, Chernick M, Rittschof D, Hinton DE. Chronic Dietary Exposure to Polystyrene Microplastics in Maturing Japanese Medaka (Oryzias latipes). Aquatic Toxicology. 2019;in press.

10. Vandenberg LN, Colborn T, Hayes TB, Heindel JJ, Jacobs DR, Jr., Lee D-H, et al. Hormones and endocrine-disrupting chemicals: Low-dose effects and nonmonotonic dose responses. Endocrine Reviews. 2012;33(3):378-455. doi: 10.1210/er.2011-1050.

11. Kortenkamp A. Ten years of mixing cocktails: A review of combination effects of endocrine-disrupting chemicals. Environ Health Perspect. 2007;115:98-105. doi: 10.1289/ehp.9357. PubMed PMID: WOS:000207172600015.

12. Avagyan R, Luongo G, Thorsén G, Östman C. Benzothiazole, benzotriazole, and their derivates in clothing textiles—a potential source of environmental pollutants and human exposure. Environ Sci Pollut Res. 2015;22(8):5842-9. doi: 10.1007/s11356-014-3691-0.

13. Tangtian H, Bo L, Wenhua L, Shin PKS, Wu RSS. Estrogenic potential of benzotriazole on marine medaka (Oryzias melastigma). Ecotoxicology and Environmental Safety. 2012;80:327-32. doi: 10.1016/j.ecoenv.2012.03.020.

How about the possibility for loading of other chemical contaminants into the body together with microfibers ingestion in a natural situation? Addition of this information would strengthen the quality of this manuscript.

Response: Thank you for your suggestion. 

Several dyes and chemicals used in the manufacture of textiles have been shown to be acutely toxic [1, 2] or carcinogenic [3]. However, there are knowledge gaps as to MF behavior in the environment [4]. The concepts of capacity, delivery, and resorption of various compounds are appearing with increasing frequency in the literature. Adding to the complexity of microplastic chemistry is predicting and interpreting sorption of metals, flame retardants, plasticizers, organic pollutants, and other compounds to plastics in the environment [5-9]. Moreover, plastics have the capacity to sorp organic molecules with increasing lipophilicity [10, 11], a property with potentially large biological implications. Once in the environment and following ingestion, dyes and additives can leach [12] and any sorped compounds can desorp [13, 14] within the digestive tract. Teasing apart effects of sorped contaminants in addition to mechanical damages caused by particles and physiological changes from plastic additives is extremely complex. For this reason, we emphasize the need to include pristine plastic controls in future studies investigating effects of other contaminants.

We have integrated this information into the Discussion (TC464-477; M446-459).

1. Athira N, Jaya DS. The use of fish biomarkers for assessing textile effluent contamination of aquatic ecosystems: a review. Nature Environment and Pollution Technology. 2018;17(1):25-34. PubMed PMID: 2016346742.

2. Selvaraj D, Leena R, Kamal D. Toxicological and histopathological impacts of textile dyeing industry effluent on a selected teleost fish Poecilia reticulata. Asian Journal of Pharmacology and Toxicology. 2015;3(10):26-30.

3. Lithner D, Larsson Å, Dave G. Environmental and health hazard ranking and assessment of plastic polymers based on chemical composition. Science of The Total Environment. 2011;409(18):3309-24. doi: 10.1016/j.scitotenv.2011.04.038.

4. Barrows APW, Cathey SE, Petersen CW. Marine environment microfiber contamination: Global patterns and the diversity of microparticle origins. Environmental Pollution. 2018;237:275-84. doi: 10.1016/j.envpol.2018.02.062.

5. Brennecke D, Duarte B, Paiva F, Caçador I, Canning-Clode J. Microplastics as vector for heavy metal contamination from the marine environment. Estuarine, Coastal and Shelf Science. 2016;178:189-95. doi: 10.1016/j.ecss.2015.12.003.

6. Rochman CM. The complex mixture, fate and toxicity of chemicals associated with plastic debris in the marine environment. In: Bergmann M, Gutow L, Klages M, editors. Marine Anthropogenic Litter. Cham: Springer International Publishing; 2015. p. 117-40.

7. Rochman CM, Hoh E, Hentschel BT, Kaye S. Long-term field measurement of sorption of organic contaminants to five types of plastic pellets: Implications for plastic marine debris. Environmental Science & Technology. 2013;47(3):1646-54. doi: 10.1021/es303700s.

8. Turner A, Holmes LA. Adsorption of trace metals by microplastic pellets in fresh water. Environmental Chemistry. 2015;12(5):600-10. doi: 10.1071/EN14143.

9. Bakir A, Rowland SJ, Thompson RC. Competitive sorption of persistent organic pollutants onto microplastics in the marine environment. Marine Pollution Bulletin. 2012;64(12):2782-9. doi: 10.1016/j.marpolbul.2012.09.010.

10. Lee H, Shim WJ, Kwon J-H. Sorption capacity of plastic debris for hydrophobic organic chemicals. Science of The Total Environment. 2014;470-471:1545-52. doi: 10.1016/j.scitotenv.2013.08.023.

11. Hüffer T, Hofmann T. Sorption of non-polar organic compounds by micro-sized plastic particles in aqueous solution. Environmental Pollution. 2016;214:194-201. doi: 10.1016/j.envpol.2016.04.018.

12. Suhrhoff TJ, Scholz-Böttcher BM. Qualitative impact of salinity, UV radiation and turbulence on leaching of organic plastic additives from four common plastics — A lab experiment. Marine Pollution Bulletin. 2016;102(1):84-94. doi: 10.1016/j.marpolbul.2015.11.054.

13. Turner A. Mobilisation kinetics of hazardous elements in marine plastics subject to an avian physiologically-based extraction test. Environmental Pollution. 2018;236:1020-6. doi: 10.1016/j.envpol.2018.01.023.

14. Anbumani S, Kakkar P. Ecotoxicological effects of microplastics on biota: a review. Environ Sci Pollut Res. 2018;25(15):14373–96. doi: 10.1007/s11356-018-1999-x.

---

## [Decision Letter · Decision Letter 1]

19 Feb 2020

Chronic microfiber exposure in adult Japanese medaka (Oryzias latipes)

PONE-D-19-29596R1

Dear Dr. Hinton,

We are pleased to inform you that your manuscript has been judged scientifically suitable for publication and will be formally accepted for publication once it complies with all outstanding technical requirements.

With kind regards,

Aldo Corriero, Ph.D.

Academic Editor

PLOS ONE

Additional Editor Comments (optional):

Thank you for your thoughtful consideration of all the reviewers' and Editors' comments and congratulations for the high quality of the revised manuscript.

Reviewers' comments:

Reviewer's Responses to Questions

**Comments to the Author**

1. If the authors have adequately addressed your comments raised in a previous round of review and you feel that this manuscript is now acceptable for publication, you may indicate that here to bypass the “Comments to the Author” section, enter your conflict of interest statement in the “Confidential to Editor” section, and submit your "Accept" recommendation.

Reviewer #1: All comments have been addressed

Reviewer #2: All comments have been addressed

2. Is the manuscript technically sound, and do the data support the conclusions?

Reviewer #1: Yes

Reviewer #2: Yes

3. Has the statistical analysis been performed appropriately and rigorously? 

Reviewer #1: Yes

Reviewer #2: Yes

4. Have the authors made all data underlying the findings in their manuscript fully available?

Reviewer #1: Yes

Reviewer #2: Yes

5. Is the manuscript presented in an intelligible fashion and written in standard English?

Reviewer #1: Yes

Reviewer #2: (No Response)

6. Review Comments to the Author

Reviewer #1: I'm happy with the reviewers decision to use PES, but I just want to note that PET, polyethlyene terephthalate and polyester are synonymous.

Reviewer #2: Authors have revised the manuscript very carefully and elegantly. I do not have any further comments.

7. PLOS authors have the option to publish the peer review history of their article (what does this mean?). If published, this will include your full peer review and any attached files.

Reviewer #1: No

Reviewer #2: No

---

## [Editor Report · Acceptance letter]

26 Feb 2020

PONE-D-19-29596R1 

Chronic microfiber exposure in adult Japanese medaka (*Oryzias latipes*) 

Dear Dr. Hinton:

I am pleased to inform you that your manuscript has been deemed suitable for publication in PLOS ONE. Congratulations! Your manuscript is now with our production department. 

With kind regards,

on behalf of

Dr. Aldo Corriero 

Academic Editor

PLOS ONE